# Recent Advances in the Processing and Properties of Alumina–CNT/SiC Nanocomposites

**DOI:** 10.3390/nano9010086

**Published:** 2019-01-11

**Authors:** Ibrahim Momohjimoh, Mohamed A. Hussein, Nasser Al-Aqeeli

**Affiliations:** 1Department of Mechanical Engineering, University of Hafar Al Batin, Hafar Al Batin 39524, Saudi Arabia; imomohjimoh@uohb.edu.sa; 2Department of Mechanical Engineering, King Fahd University of Petroleum and Minerals, Dhahran 31261, Saudi Arabia; 3Center of Research Excellence in Corrosion, Research Institute, King Fahd University of Petroleum & Minerals’ (KFUPM), Dhahran 31261, Saudi Arabia; mahussein@kfupm.edu.sa

**Keywords:** alumina-based, SiC, carbon nanotube, microstructure, functional properties

## Abstract

An alumina-based nanocomposite is fabricated through the addition of secondary nanophase material to an alumina matrix to alter and tailor the properties of alumina. The addition to alumina of semi-conductive materials, such as silicon carbide (SiC), and high conductive materials, such as carbon nanotubes with a characteristic size in the nanometer range, can alter the mechanical strength, hardness, toughness, and electrical and thermal properties of alumina. This paper discusses recent advances in the synthesis of alumina–SiC and alumina-carbon nanotube (CNT) nanopowders and their consolidation using conventional and non-conventional techniques. Mechanical (hardness, fracture toughness and flexural strength) and functional (thermal and electrical) properties are discussed. The influence of the microstructure on the properties of alumina–SiC and alumina–CNT nanocomposites is discussed. Furthermore, potential applications and current related research trends are described.

## 1. Introduction

Alumina is an extensively used technical ceramic due its hardness, high temperature strength and high thermal and electrical insulating properties. However, the inherent brittleness of alumina has limited its use in structural applications, and this has motivated researchers over the years to improve the fracture toughness of alumina [1]. Niihara and colleagues in 1991 [2] proposed that the addition of SiC to alumina can increase its fracture strength and fracture toughness. A fracture strength of over 1 GPa and a fracture toughness of 4.7 MPam^1/2^ [2] were reported. The creep resistance of alumina was also found to improve upon the addition of 5 wt% SiC. Since this work, researchers have tried to improve the mechanical properties of alumina. Nanocomposites can be defined as a composite system consisting of a matrix and homogenously dispersed phase particles, fibers or nanotubes with characteristic dimensions of less than 100 nm [3].

Alumina-based nanocomposites, which are created through the addition of suitable second phase nanoparticles or fibers, can exhibit improved mechanical and functional performance compared to alumina. Several materials, such as titanium carbide (TiC) [4], zirconia (ZrO_2_) [5,6], silicon nitride (Si_4_N_3_) [7], titanium nitrite (TiN) [8], titanium oxide (TiO_2_) [9] and silicon carbide (SiC) [10,11,12], have been used to reinforce alumina. The discovery of carbon nanotubes (CNTs) by Iijima [13] has created opportunities to enhance the mechanical and functional properties of alumina. The intrinsically strong thermal and electrical properties of SiC and CNTs have rendered them potential candidates for improving the properties of alumina. The fabrication of alumina-based nanocomposites, similar to other nanocomposites, is a major challenge, as issues of nanoparticle agglomeration still occur. The inhomogeneous distribution of second phase nanoparticles in alumina may impair its excellent properties.

Carbon nanotubes, which are either single-wall carbon nanotubes (SWCNTs) or multi-wall carbon nanotubes (MWCNTs), have been identified by researchers as potential nanofillers for use in ceramics as a result of their superior mechanical properties (Young’s modulus of ~1 TPa [14] and tensile strength of 60 GPa [15]), thermal properties (6600 W/mk for an individual SWCNT and 3000 W/mk for an individual MWCNT) [16,17,18] and electrical properties (10^6^ S/m for SWCNTs and >10^5^ S/m for MWCNTs) [19,20]. CNTs have been used as reinforcement in many polycrystalline ceramics, such as Al_2_O_3_ [21], MgAl_2_O_4_ [22], ZrO_2_ [23], SiC [24], and Si_3_N_4_ [25]. Of all these ceramics, Al_2_O_3_ is the most widely used in industrial settings due to its excellent properties and broad range of applications in gas radiant burners, wear parts, the medical, automobile, aerospace, glass and metal production industries and cutting tools [26,27]. These applications require that the thermal conductivity and electrical conductivity be as high as possible to prevent the thermal shock-related failure of component parts. However, reinforcement with highly conductive materials, such as SiC and CNTs, does not significantly improve the thermal conductivity of alumina. The moderate thermal conductivity achieved for alumina-based nanocomposites is attributed to the existence of a thermal barrier interface between alumina and the second phase, impurities, porosity and various other defects that contribute to the scattering of phonons. According to Hasselmann et al. [28], thermal barrier resistance between a matrix and a dispersion is the cause of the relatively low thermal conductivity observed in composites of this nature. Significant improvements to the functional properties of alumina-based nanocomposites can be achieved through the tailoring of properties. The addition of semi-conductive materials, such as SiC, and highly conductive materials, such as CNTs in significant amounts (percolation threshold value), can significantly improve the transport properties of alumina. The homogenous dispersion of second phase conductive particles via sonication and mechanical ball milling can reduce the percolation threshold, thus leading to interconnected network formation and enhancement of thermal and electrical properties [29]. Planetary ball milling destroys Al_2_O_3_ agglomerates that normally cause large processing flaws. Furthermore, SiC particles achieve an optimal distribution during this process to prevent agglomeration. A percolation threshold as low as 5 wt% SiC can be achieved by mechanical ball milling. However, for CNT-ceramic nanocomposites, the major challenges are CNT agglomeration, insufficient densification at high CNT content and low interfacial adhesion between CNTs and ceramics [30]. The agglomeration of CNTs occurs due to their van der Waals interactions and high aspect ratios. When CNTs form lumps, they act as impurities and, therefore, have minimal or no reinforcement effects [31]. Thus, the decrease in mechanical properties observed at high CNT contents has been attributed to the occurrence of agglomeration [32].

To tailor the mechanical and functional (thermal and electrical) properties of alumina–CNT nanocomposites, critical steps must be followed. These include homogenous distribution of CNTs in the alumina matrix to avoid agglomeration resulting from van der Waals forces [33,34], a better consolidation process (e.g., spark plasma sintering (SPS)) to avoid CNT damage during processing and optimal bonding between the CNTs and alumina through the provision of interfacial compatibility, and good adhesion. CNTs are not readily dispersed in water due to the nature of their covalent bond. Therefore, surface modification of CNTs is necessary. Chemical modifications involving the use of mixtures of nitric acid and sulfuric acid are commonly used for the purification and modification of CNTs to remove metal catalysts and amorphous carbon [35]. Again, functional groups, such as hydroxyl and carboxyl groups, are equally attached to the surfaces of CNTs as a result of their treatment. Thus, hydroxyl and carboxyl groups facilitate the dispersion of CNTs in water. Several dispersion techniques, such as conventional powder processing [36], colloidal processing [37,38], sol-gel processing [39] and chemical vapor decomposition [40], have been used for the uniform distribution of CNTs in an alumina matrix. Conventional consolidation processes (hot isostatic pressing, hot pressing, etc.) have been reported [36,40,41,42,43], although these processes required high sintering temperatures to achieve sufficient densification. High-temperature sintering has a detrimental effect on CNTs. To prevent the degradation of CNTs, SPS is an ideal consolidation technique. SPS offers several advantages, including a relatively low sintering temperatures, shorter sintering periods, high densification and high heating and cooling rates [44,45].

The superior mechanical properties of alumina-based nanocomposites have been attributed to several mechanisms. The addition of SiC nanoparticles changes the mode of fracture from intergranular to transgranular as a result of grain boundary strengthening. The SiC nanoparticle is also believed to exert pinning pressure on the alumina matrix, which inhibits the grain boundary mobility of the alumina matrix [46]. This microstructural refinement with SiC addition is responsible for the exceptional hardness and strength of alumina–SiC nanocomposites [47]. Other mechanisms (e.g., crack bridging, crack deflection, micro-cracking, and elastic modulus and coefficient thermal expansion mismatch between Al_2_O_3_ and SiC) contribute immensely to the mechanical performance of alumina–SiC nanocomposites [48]. Furthermore, nanotube pull-out [49] and microstructural refinement [32] have been observed in alumina–CNT nanocomposites as major strengthening mechanisms.

The transport (thermal and electrical) properties of alumina-based nanocomposites are dependent on the content of the conductive reinforcement phase and the extent of its dispersion. The interfacial resistance of nanocomposites is very high due to the high surface area of nanoparticles. Several models, such as Hasselman and Johnson’s model, Maxell’s model and Hamilton and Crosser’s model [28], have been developed to estimate the thermal conductivity of ceramic composites. However, none of these models can be used to evaluate the exact thermal conductivity of nanocomposite materials, as the interfacial resistance is difficult to measure [50]. Thermal conductivity (λ) is expressed as a function of the thermal diffusivity (α), specific heat capacity (*C*p) and bulk density (ρ) of nanocomposites [51].
(1)$${\lambda =\alpha \rho C}_{p}$$

Thermal diffusivity is typically determined by the flash method. This method involves the uniform radiation of a small specimen with a short pulse of energy. A short measurement time, a completely non-destructive nature and accurate reproducibility of the results are the features of this method [51].

The electrical conductivity of nanocomposites increases with increasing content of the conductive reinforcement phase, as numerous conductive paths with substantial charge carrier mobility are available. Moreover, the conductivity increases with measurement temperature, as the carrier mobility increases with increasing temperatures. Electrical conduction has been described in terms of a model called fluctuation induced tunneling (FIT), whereby an electron is conceived to tunnel through an insulating barrier. The miniaturization of photonic and electric devices has called for the development of a material with improved electrical conductivity through the reduction of the percolation threshold. The addition of nanosized particles to alumina ceramics can generate more points of contact due to the high surface areas, resulting in high resistivity relative to micro-sized particles. The electrical resistance is defined as [51],
(2)$$R\;=\;\frac{V}{I}$$
where *V* is the voltage across the surface and *I* is the applied current.

The resistance evaluated can be used to measure the electrical resistivity (ρ) of the material [51],
(3)$$\rho \;=\;\frac{RA}{t}$$
where *A* is the cross-sectional area and *t* is the thickness of the material. Electrical conductivity can, therefore, be measured using the following expression [52]:(4)$$\sigma \;=\;\frac{1}{\rho }$$

In this review article, two nanocomposites (alumina–SiC and alumina–CNT nanocomposites) are discussed in terms of their preparation, mechanical properties, microstructures, thermal properties and electrical properties, and applications of the nanocomposite system are elucidated.

## 2. Processing of SiC–Carbon Nanotube (CNT)/Alumina Nanocomposites

### 2.1. Powder Processing

Powder processing is the first step of ceramic nanocomposite processing, and it involves the widest variety of manufacturing technologies due to its ability to economically produce and fabricate high-quality complex components from many materials close to their tolerances [53]. The high aspect ratio of nanopowders [54] is an important characteristics that leads to behavior that lies between that of solids and that of fluids. Various techniques are used in the preparation of nanopowders, including plasma processing, reactive synthesis, mechanical alloying [55], chemical precipitation [56], gas atomization [57], sol-gel processing [39], sonication [58,59], and molecular level mixing [60]. Among the techniques, mechanical alloying is the most widely used for ceramic nanopowder processing, as this process is simple to apply and requires little technical skill to operate.

### 2.2. Mechanical Alloying

Mechanical alloying is a technique used for powder preparation that allows for the homogenous synthesis of materials from a constituent powder mixture. The technique was invented by John Benjamin and colleague in 1966 at the research laboratory of the International Nickel Company (INC) as a result of their search for a nickel-based superalloy with elevated temperature, mechanical strength, and corrosion and oxidation resistance for application in gas turbine engines [61]. The powder preparation technique involves repeated cold welding, fracturing, and re-welding of elemental powder particles in a high-energy ball mill. In the literature, mechanical alloying (MA) and mechanical milling (MM) are two different terms normally used interchangeably to describe the synthesis of powder particles in high-energy ball mills. MA involves milling of a powder mixture through the transfer of material to obtain a homogenous powder mixture, while mechanical milling describes a milling process in which no materials are used for the homogenization of a powder mixture [62].

The MA method begins with the mixing of powder particles in the right composition and then loading the powder mixture into a milling container (vials) along with a grinding medium (milling balls). The mixture is milled for a given length of time until a steady state is achieved when the composition of the powder particles is almost the same as that in the starting powder mixture. The milling time is dependent on several factors, such as the rotational speed, ball-to-powder ratio, initial powder crystallite size and milled material hardness [63]. Key elements of MA thus include the powder raw materials, milling devices and process variables. Different types of mechanical milling machines are available, and specifications of the milling process and milling variables are contained in the paper [61].

### 2.3. Ultrasonication

Ultrasonication is the process of supplying ultrasound energy to stir particles in a solution. This is achieved in the laboratory by using an ultrasonic bath or an ultrasonic probe, which are also called sonicators [58]. Ultrasonication is the most frequently used technique for the dispersion of nanoparticles. During ultrasonication, ultrasound is propagated through a series of compression and attenuated waves, which are transferred to the molecules of the liquid medium as they propagate. The shock waves produced assist in the peeling off of individual nanoparticles, and this results in the splitting of individualized nanoparticles from bundles or agglomerates. Standard laboratory sonicators operate at 20–23 kHz with a power of typically less than 100 W. However, commercial sonicators operate at a power ranging from 100 to 1500 W with adjustable amplitudes of 20% to 70%. The probes are fixed to a base unit that tapers down to a tip with a diameter of 1.6 to 12.7 mm. This clearly shows that the power generated from the wide base is concentrated at the tip, thus affording the probe a high degree of energy intensity. Sonication is suitable for the dispersion of Al_2_O_3_ and SiC nanoparticles and carbon nanotubes (CNTs) in low viscous liquids, such as water, acetone, and ethanol [13].

### 2.4. Powder Consolidation

Consolidation of powder consists of the assembly of powder particles into a specific product to attain desired geometries, structures or properties. Consolidation processes are dependent on the application of mechanical, chemical and thermal energy to realize bonding and consolidation of powder particles. Nanopowder consolidation is an essential component of densification with minimal microstructural transformation. In most cases, densification typically results in either grain coarsening or the formation of specimens that are too small and exhibit insufficient bonding capacities. This has a detrimental effect on nanomaterial properties, especially on mechanical properties. The need to retain nanofeatures in consolidated nanomaterials has caused researchers and scientists to develop more efficient consolidation processes. The consolidation techniques have a direct influence on the nanomaterial microstructure, and to explore the interesting properties of nanomaterials, efficient consolidation techniques are needed [64,65]. Consolidation is broadly classified into two types: conventional and non-conventional consolidation. In conventional consolidation processes, heat energy is produced externally and transferred to the powder through various heat transfer processes, such as convection, conduction and radiation, while in nonconventional consolidation, materials either absorb microwave energy to generate heat within themselves or electrical discharge resistance heating and pressure are applied to effect the sintering of powder particles [55]. Conventional powder sintering techniques include pressureless sintering, hot pressing (HP), hot isostatic pressing (HIP) and cold isostatic pressing (CIP). Pressureless sintering [66] is not commonly used for nanopowder consolidation, as the process occurs in the absence of pressure but at high sintering temperatures. Grain growth leads to the loss of nanofeatures. Hot pressing involves a combination of uniaxial pressure and temperature to enhance the bonding of nanopowders. Better properties can be achieved using this method [67,68]. Hot isostatic pressing combines temperature and pressure in the consolidation of nanopowders to effect sintering; however, HIP is different from HP, as pressure is applied in all directions in the former case [69]. This eliminates the anisotropy in the properties of composites, as might occur for HP consolidated materials. Cold isostatic pressing is similar to HIP except that heat is not involved. Non-conventional consolidation is the best means of sintering nanopowders, as the process occurs at relatively low temperatures. Grain growth problems are in turn mitigated, and nanofeatures in the powder are retained in bulk nanocomposites. Spark plasma sintering (SPS) [70] and microwave sintering (MS) [71] are two commonly used non-conventional consolidation techniques. In MS, the microwave energy generated is absorbed by powder materials and then transformed into heat sufficient to consolidate powders [72]. This involves the conversion of electromagnetic radiation energy into thermal energy. In some cases, hybrid microwave sintering is employed for the sintering of nanopowders. Hybrid sintering in this case involves a combination of microwave heating and pressure to effect consolidation [73]. Ghasali et al. [74] studied the distinction between microwave and spark plasma sintering of Al-B4C composites and discovered that spark plasma samples exhibit higher mechanical properties than the microwave sintering counterparts. The microwave sintering technique is rarely used for the consolidation of alumina-based nanocomposites, probably because of the dielectric properties of alumina.

#### Spark Plasma Sintering (SPS)

Spark plasma sintering (SPS), which is also known as plasma activated sintering (PAS), is a novel field-assisted sintering technique that uses uniaxial forces and pulsed (on-off) direct electrical currents at low temperatures and pressure for powder consolidation [75]. Spark plasma sintering offers several advantages (i.e., high heating rates, high cooling rates, relatively low sintering temperatures, high densification, short holding periods and an absence of pre-compaction or the use of binders [76]), rendering SPS the best sintering technique. SPS operational procedures start with the creation of a vacuum to prevent the thermal oxidation of the powder. This is immediately followed by pressing and then heating to sintering temperatures. In a third stage, the compressed powder is held at the sintering temperature for a given length of time. Finally, the system cools to ambient temperature. Parameters that control the temperature distribution in the SPS process are the electrical conductivity of the powder material, the thickness of the die wall and the availability of the graphite sheet used to prevent direct contact from occurring between the graphite mold and specimen [77]. The mechanisms and principles of SPS processes are centered on the electrical spark discharge phenomenon, in which a high energy with minimum voltage spark pulse current produces spark plasma at high confined temperatures between particles, resulting in optimal thermal and electrolytic diffusion. Most SPS temperatures range from low to approximately 2000 °C, which is 200 and 500 °C below those of conventional available furnaces. According to Kessel et al. [78], the mechanism of the SPS process involves the ionization of particle in contact due to spark discharge, which generates impulsive forces that ease the diffusion of atoms at points of contact. In their work Guillon et al. [79] claimed that a pulsed current cleans particle surfaces as a result of their observation that an interface was formed between particles without oxidation. However, the subject of plasma generation has not yet been directly verified from experiments. Vanmeensel et al. [80] explained that joule heating as a result of the flow of electric currents through particles favors the bonding of particles under mechanical forces. The intensity of the heat generated often causes the material to reach a boiling point, enhancing the localized vaporization or cleaning of powder surfaces. It must be noted that in SPS processes, powder surfaces are easier to clean and activate compared to conventional electrical sintering techniques since the technique promotes micro and macro levels of material transfer. Therefore, a consolidated mass of superior quality is usually generated at low sintering temperatures and over shorter periods of time than through the conventional consolidation process. Figure 1 shows the SPS configurations.

### 2.5. Processing of Alumina–SiC Nanocomposites

There are several means of processing Al_2_O_3_–SiC nanopowders, including conventional powder preparation, sol-gel synthesis, and polymer processing, and all these methods are aimed at the homogenous distribution of nano-SiC particles into the alumina matrix. Conventional powder processing is the most popular means of alumina–SiC nanocomposite processing and involves raw material selection, dispersion of powder in a liquid medium and slurry drying [48]. The raw materials are alumina and SiC of high purity, and a small grain size and high purity of the powders are required to prevent the formation of a secondary phase during sintering. In their work, Gao et al. [81] prepared 5 vol% SiC-Al_2_O_3_ powder using Nano-SiC particles, aluminum chloride, and ammonia. The nano-SiC particles were ultrasonically dispersed to break agglomerates, and the pH was carefully adjusted to between 9 and 10. The suspended aqueous nano-SiC was continuously stirred at room temperature, and then, a solution of aluminum chloride and ammonia was added and continuously stirred until complete precipitation occurred. The SiC-Al_2_O_3_ gel that formed was rinsed with distilled water until it was free of chloride ions and then dried at 100 °C. Thereafter, the gel was calcined at 700 °C and then sieved through a 200 mesh sieve to obtain the final SiC-Al_2_O_3_ powder. Wang et al. [82] synthesized 5% and 10 vol% SiC-Al_2_O_3_ powder using a combination of sol-gel and wet ball milling techniques. For this preparation, AlCl_3_ was added to SiC particles, and the mixture was then stirred. Ammonium hydroxide (NH_4_OH) was then added until precipitation occurred. The resulting SiC-Al_2_O_3_ powder was obtained after drying, calcining and attrition milling of the slurry.

Al_2_O_3_–SiC powder preparation via sonication and mechanical ball milling using zirconia balls and media has been reported by several authors [12,83,84]. For these preparations, SiC powder was dispersed in distilled water together with alumina powder, and the mixture was sonicated for 20 min. The slurry was then transferred to an attritor mill with zirconia balls and media and then, the balls were milled for 2 h at a speed of 500 rpm. A pH of 9 was maintained for the dispersion, the slurry was dried for 24 h, and finally, the balls were milled for 1 h. Al_2_O_3_–SiC powder was obtained by sieving through a 150 µm sieve. Parchoviansky and co-workers [85] prepared Al_2_O_3_/SiC nanopowders containing different fractions of SiC via conventional mixing of alumina powder with nanosized silicon carbide powders. Here, the powder mixtures were uniformly dispersed by ball milling in a medium containing isopropyl alcohol for 24 h in a polyethylene flask using high purity Al_2_O_3_ balls and a powder-to-ball ratio of 1:10. The slurry was evaporated in a vacuum oven, and the dried soft agglomerated nanopowders were then crushed in an agate mortar and pestle. The detailed preparation of Al_2_O_3_–SiC powder is discussed in Borrell et al. [86]. A powder mixture containing 17 vol% nano-SiC was ball milled using ethanol as a solvent. High purity (99.5%) alpha alumina balls 2 mm in diameter were added to the media at a ball-to-powder ratio of 4:1. Milling was performed at a speed of 100 rpm for 48 h, and the slurry was dried at 60 °C to obtain a soft agglomerate. The agglomerate was crushed in a mortar and pestle and sieved through a 60 µm sieve. The preparation of Al_2_O_3_–SiC nanopowders using alumina milling balls and milling vials has been reported in several papers [10,87,88,89,90,91]. The powder mixture in isopropyl ethanol was ball milled using alumina balls in alumina media for 24 h at a speed of 100 rpm. The powder mixture was dried in an oven for 24 h, and the agglomerated dried powder that formed was ball milled for 12 h and sieved through sieves of appropriate sizes to obtain nanopowders for consolidation.

As reported by Liu et al. [92] and Watanabe and Kimura [93], the combination of sonication and planetary ball milling techniques using alumina balls and vials has been found to be very effective for the dispersion of SiC nanoparticles into an alumina matrix. In their work, a mixture of SiC and alumina powder in ethanol was sonicated for 45 min, and the resulting slurry was then transferred into alumina vials containing alumina balls and then ball milled for 2 h. The powder mixture was dried and crushed in a mortar and pestle and was then sieved prior to consolidation. The use of ultrasonication and magnetic stirring for the preparation of Al_2_O_3_–SiC powder was reported by Sciti et al. [94]. In their work, a powder mixture containing Al_2_O_3_–SiC powder was magnetically stirred, followed by sonication for 2 h. The slurry was then dried and sieved to obtain Al_2_O_3_–SiC nanopowder. Al_2_O_3_–SiC nanopowder has been prepared by Ko et al. [95] using SiC balls and SiC media. A mixture of alumina and SiC in ethanol was ball milled for 24 h using SiC balls in a polyethylene jar. The slurry was then dried and sieved through a 60 µm mesh. It should be noted that powder preparation is vital to nanocomposite processing, as the properties of nanocomposites depend heavily on how well the powder is prepared.

All the techniques described above are designed to ensure that SiC particles are dispersed homogenously in an alumina powder matrix. A combination of techniques, such as sonication and ball milling, has been found to be effective in the homogenous distribution of SiC in an alumina matrix. The proper control of sonication and ball milling parameters is paramount to achieving a homogenous distribution of SiC in an alumina matrix.

### 2.6. Processing of Alumina–CNT Nanocomposites

CNTs are classified as SWCNTs or MWCNTs, depending on the rolling layers of graphene sheets. Despite the excellent properties of CNTs, their reinforcing effects on ceramic matrices have not been fully realized, as they are difficult to disperse and are not compatible with the ceramic matrix because of their bonding nature and hydrophobicity. There are several techniques for the production of CNTs, including arc discharge [96], laser ablation [97] and chemical vapor deposition [98]. CNTs by their design are chemically unreactive in pristine form, which is typically not desirable for certain applications. To overcome this challenge, it is necessary to functionalize CNTs. The process of functionalization improves their dispersion and consequently their application potential.

#### Functionalization of CNTs

The addition of oxygen-containing species to the surfaces of CNTs promotes their solubility in aqueous or organic solvents and decreases van der Waals associations between different CNTs, enhancing the dissociation of nanotube bundles into individual tubes. There are essentially two means of functionalizing CNTs: chemical and physical functionalization [99,100]. Chemical functionalization occurs as a result of the covalent linkage of functional entities to carbon scaffolds of CNTs. Chemical functionalization is typically executed using strong acids, such as HNO_3_, H_2_SO_4_ or a mixture of them [101], or using strong oxidants, such as KMnO_4_ [99]. The treatment process involves employing a mixture of CNTs, nitric acid and sulfuric acid of high concentration, followed by 3 h of sonication. The mixture is then stirred for 24 h and rinsed with distilled water until the pH is 7. The CNTs are sieved and dried overnight at 100 °C [66]. Defects created by the oxidation agents are neutralized through their bonding with carboxylic acid (-COOH) or hydroxyl (-OH) groups. These functional groups have sufficient chemistry, and CNTs can be used as a precursor for further chemical reactions. Mansoor et al. [102] investigated two chemical methods of functionalization that involve using nitric acid and hydrogen peroxide, and they found that hydrogen peroxide-based functionalization is more effective than nitric acid functionalization. The MWCNTs were found to be completely de-roped in hydrogen peroxide, and better dispersion outcomes were observed in the epoxy, resulting in high mechanical properties of MWCNT-epoxy nanocomposites.

The physical functionalization of carbon nanotubes involves the adsorption of surfactant molecules on the surfaces of CNTs to ensure that agglomerations do not form. Several factors, such as the characteristic properties of the surfactant, the chemistry of the medium and the nature of the polymer matrix, determine the efficiency of the physical functionalization of CNTs. However, two major drawbacks of the chemical and physical functionalization of CNTs include damage generated as a result of ultrasonication or ball milling [103]. In some cases, such damage can occur on the sidewalls of CNTs, which can result in the severe degradation of the mechanical and functional performance of CNTs [100]. Benefits of CNT functionalization include the removal of impurities such as catalyst remnants, the addition of hydroxyl or carboxyl groups for bonding with the matrix and high levels of dispersion in water [13].

The dispersion of CNTs in an alumina matrix is crucial to the processing and fabrication of alumina–CNT nanocomposites. The aggregation of CNTs during the processing of nanocomposites is a major constraint that limits the exploitation of their potential. CNTs by nature are chemically inert due to unique SP2 bonds that form in their graphene layers and the complex entanglements formed by strong van der Waals forces [104]. Well-dispersed CNTs in alumina can lead to the production of high-quality nanocomposites with a higher density and strong mechanical and functional properties [105]. Mechanical methods, such as sonication and ball milling, have been used to disperse CNTs in an alumina matrix effectively. Sonication produces ultrasonic waves that are most effective in dispersing CNTs by disrupting the van der Waals attractive forces that exist among the cluster tube by generating a shear force in the liquid medium [106]. A modified sonication technique known as gas-purging sonication, recently developed by Bakhsh et al. [107] uses a combination of nitrogen gas and sonication to ensure a better dispersion of CNTs in a liquid medium. The high-energy ball milling of Al_2_O_3_-CNT for 12 to 24 h using Al_2_O_3_ or ZrO_2_ balls has also been reported. Milling has been performed to reduce the particle size of the starting Al_2_O_3_ and to promote de-agglomeration of nanosized particles [108,109,110]. Other researchers have reported the use of sol-gel and chemical vapor deposition for the dispersion of CNTs in alumina. For instance, Zhang et al. [111] and Kumari et al. [112] directly dispersed CNTs in alumina by growing CNTs on alumina nanoparticles using Co(NO_3_)_2_·6H_2_O as a catalyst, and the mixture of alumina and CNTs in ethanol was sonicated for 15 min. The mixture was dried overnight at 130 °C and then ground into powder.

Alumina–CNT nanopowders have been synthesized by Barzegar-Bafrooei and Ebaddzadeh [53] using the sol-gel method. In their study, a homogenous mixture of carbon nanotubes and alumina particles was obtained through the dispersion of carbon nanotubes and alumina in a solvent, followed by heating into a gel. The resultant gel formed was evaporated and calcined at 200 °C into boehmite-CNT nanocomposite powders. Sonication and planetary ball milling processes for the dispersion of CNT in alumina have been reported by some authors [113,114,115]. In this process, CNTs are dispersed in dimethylformamide (DMF) through the operation of high power sonicators for 2 h and then hand mixed with alumina nanopowders. The mixture is ball milled for 8 h, dried for 12 h at 75 °C on a heating plate in air, and then oven dried at 100 °C for 3 days. Soft agglomerates are crushed and sieved to obtain alumina–CNT nanopowders. Molecular level mixing [60,116] approaches have been employed for the synthesis of alumina–CNT powders, and the homogenous distribution of CNTs in alumina has been reported with consequent improvement in the mechanical properties. Recently, Fawad and colleagues [104] studied the effects of dispersants on the properties of alumina–CNT nanocomposites. Three different dispersants (gum arabic (GA), sodium dodecyl sulphate (SDS) and water) were used to disperse CNTs in an alumina matrix using an ultrasonic bath for 6 h, followed by ball milling for 8 h. The slurry was finally dried at 80 °C for 12 h. The results confirm that the combination of GA and SDS is the most effective approach, followed by the use of GA alone. The use of SDS and water was found to disperse CNTs in alumina the least. In conclusion, conventional powder processing involving the use of ultrasonication and ball milling and molecular level mixing is suitable for dispersing CNTs in a ceramic matrix at low concentrations of CNTs. However, for higher volume fractions of CNTs in an alumina matrix, the use of colloidal heterocoagulation is highly recommended for their homogenous distribution.

### 2.7. Consolidation of Alumina/SiC and Alumina–CNT Nanopowders

Consolidation of nanopowders is the next step following powder preparation. Advances in technology have brought about significant progress in practices and theories of nanosintering, resulting in the manufacture and fabrication of dense parts with a nanometer grain size [2,64]. The driving factor for sintering of nanopowders is their high surface area. Nanopowders are highly unstable due to their high surface energy. The tendency to reduce their high surface energy is the driving factor behind their consolidation [117]. Wang et al. [82] consolidated alumina–SiC nanopowders through hot pressing at a pressure of 35 MPa and at sintering temperatures of 1650 °C and 1750 °C. Higher densification was achieved at 1650 °C and 35 MPa, but a further increase in temperature to 1750 °C did not lead to a change in densification, though the microstructural analysis showed significant grain growth and a loss of nanostructures (grain size >100 nm). Spark plasma sintering of Al_2_O_3_–SiC nanopowders is reported in several papers [81,86,90,92,118,119,120,121] for different sintering temperatures, dwelling periods, heating rates and applied pressure. Borrell et al. [86] consolidated Al_2_O_3_-17 vol% SiC nanopowders by SPS by inserting a composite powder mixture into a graphite die 20 mm in diameter and then applying uniaxial cold pressure at 30 MPa. The compressed powders were sintered at 1400 and 1550 °C for a dwelling period of 1 min and a heating rate of 100 °C/min. The sintering pressure was maintained at 16 MPa, was increased while heating to 80 MPa over the next 100 °C and remained constant throughout the holding period. The grain size of the alumina was found to be smaller at a sintering temperature of 1400 °C (430 nm) than that of the composite sintered at 1550 °C (590 nm); a meaningful change occurred in the densification of the composites sintered at the two different temperatures. Conventional consolidation techniques, such as hot pressing [2,11,85,87,89,94,122,123,124], cold isostatic pressing (CIP) [83,84,125], and hot isostatic pressing (HIP) [91,126], have been equally used for the consolidation of Al_2_O_3_–SiC powder. High sintering temperatures, long sintering periods and a lack of proper control over grain sizes limit the application of conventional sintering in the processing of nanocomposites. The two sintering techniques most widely used for alumina–CNT nanopowders are HP and SPS. HP and SPS differ in how thermal energy is transferred to powders. In the case of HP, heat is transferred by convection/radiation through graphite heating elements surrounding the pressing tool to the powder. In contrast, the SPS process involves flowing electric currents through graphite punches and dies such that the pressing tool acts as a heating element via the Joule effect. Again, the heating rate of HP is limited to 20 °C/min, while that of SPS reaches a value of 1000 °C/min. Although SPS has a high heating rate that saves time and energy, such a high heating rate can produce an inhomogeneous microstructure, large grains and low density [127]. The consolidation of alumina–CNT powder by HP and SPS was studied by Yazdani et al. [127]. Higher grain growth was observed in SPS samples than in HP samples, especially for monolithic alumina. The authors concluded that the high heating rate of SPS results in inhomogeneous heat distribution and that this effect is more pronounced when electrically sintering insulating materials, such as Al_2_O_3_, via SPS. However, the addition of electrically conductive nanophases, such as CNTs, can improve the conductivity, thereby changing the heat distribution and microstructure of the nanocomposite consolidated by SPS. Frankly speaking, consolidation of alumina–CNT nanopowders is mainly performed via SPS [21,111,115,128] and HP [43,129,130] techniques, and only in few cases has pressureless sintering [131] been used. This is likely attributable to the fact that pressureless sintering is applied at relatively high temperatures compared to the HP and SPS techniques, and the use of such extreme temperatures can result in grain growth and in the destruction of CNTs. Thus, HP and SPS are the most suitable techniques for the sintering of alumina–CNT nanopowders. For the sintering of monolithic alumina or alumina with low concentrations of CNTs, the use of HP is highly recommended, while for the sintering of alumina with high volume fractions of CNTs, SPS is the most suitable technique.

### 2.8. Densification of Al_2_O_3_–SiC Nanocomposites

Pressureless sintering techniques used for the consolidation of Al_2_O_3_–SiC nanopowders do not increase densification. Typically, as a result of composite powder preparation, SiC particles are positioned at the interfaces between alumina matrix grains, as they serve as an obstacle to densification that restricts grain boundary motion through the pinning effect. The use of pressure assisted sintering techniques, such as hot pressing, hot isostatic pressing, and spark plasma sintering, can produce composites of full densification. To achieve higher densification in pressureless sintering, a high sintering temperature is required, which promotes grain growth that impairs the mechanical and functional properties of the nanocomposites [66]. To reduce sintering temperatures during pressureless sintering, additives are usually added to aid sintering at relatively low temperatures. For the pressureless sintering of Al_2_O_3_–SiC, the powder is doped with 0.1 wt% MgO and Y_2_O_3_ to reduce the sintering temperature from 1800 to 1300 °C [126] while maintaining a coarse microstructure.

The successful densification of nanocomposites by pressureless sintering is limited by the content of SiC nanoparticles, which normally does not exceed a value of 10 vol%. Full densification of composites with a relatively high SiC content (>20 vol%) can be achieved through the application of pressureless sintering and hot isostatic pressure [48,132]. Recently, Saheb and Mohammad [133] reported higher relative density (99.3% for alumina containing 5 wt% SiC and 99.03% for alumina containing 10 wt% SiC) and monolithic alumina (99.3%) after SPS consolidation at 1500 °C and 50 MPa for 10 min of holding and a heating rate of 100 K/min. The authors explained that local elevated temperatures produced because of the spark discharge across the gap between powder particles was responsible for the high densification of pure alumina and nanocomposites. Externally applied pressure also contributed significantly to densification, as the application of pressure results in the rearrangement of particles and in the breakdown of agglomerates. In conclusion, SPS has been adjudged to increase densification, with better mechanical and functional properties compared to those achieved using conventional sintering techniques.

### 2.9. Densification of Al_2_O_3_-CNT Nanocomposites

Similar to SiC particles, CNTs have a pinning effect on the alumina matrix that resists densification. Specifically, their high aspect ratio, high specific surface area and incompatibility with the surrounding matrix cause CNTs to behave in two distinct ways: (1) reducing the sintered density and (2) decreasing the size of alumina due to a reduction in grain boundary migration through efficient pinning [23,58]. The presence of CNT clusters at grain boundaries as a result of imperfect de-agglomeration also contributes to a reduction in density. The agglomerates act as obstacles at the boundary interfaces that resist densification. Thus, increasing the content of CNTs in the nanocomposite increases the difficulty of densification of CNT-containing nanocomposites. While several attempts have been made to sinter alumina–CNT nanopowders, these attempts have been unsuccessful, as the high sintering temperatures caused oxidation of CNTs in ceramics. Zhang et al. [134] discovered that the relative density of alumina–CNT nanocomposite decreases from 98.5% to 95% when the CNT content of alumina increases from 1 to 3 wt%. Again, to prevent the oxidation of CNTs, the maximum sintering temperature of alumina–CNT nanopowders is usually kept at less than 1550 °C [107]. To accomplish this, more attention has been paid to pressure assisted sintering techniques. SPS has been considered by researchers to be an ideal sintering technique for the consolidation of alumina–CNT nanopowders due to its high heating rate, relatively low sintering temperature and shorter sintering period. This approach results in the production of a fine microstructure with enhanced mechanical and functional performance. Saheb and Hayat [135] found a 99.8% relative density for alumina after SPS consolidation at 1500 °C and a pressure of 50 MPa. The relative density decreased to 97% and 95% when 1 and 2 wt% MWCNTs, respectively, were added to alumina. Thus, CNTs are believed to have a pinning effect on the alumina matrix that prevents grain boundary sliding and thus inhibits densification. The combination of sintering pressure and joule heating offered by SPS has made it a central technique for the sintering of alumina–CNT nanocomposites with improved mechanical and functional performance.

## 3. Mechanical Properties of Alumina-Based Nanocomposites

### 3.1. Mechanical Properties of Alumina–SiC Nanocomposites

The mechanical properties of alumina–SiC composite systems were first reported by Niihara [2] after the addition of 5 wt% SiC to alumina. According to Niihara, a strength of 1005 MPa was achieved for nanocomposites, while that achieved for monolithic alumina was 380 MPa. The fracture toughness was also found to improve by 40%. Niihara concluded that this increase in strength was a result of an overall refinement of the microstructure and a reduction in processing flaws resulting from the addition of fine SiC particles to the alumina matrix [2,7,123]. Several researchers have consolidated alumina–SiC nanocomposites with different volumes of SiC nanoparticles, but none have reproduced the results reported by Niihara. For example, Gao et al. [81] investigated the mechanical properties and microstructures of alumina–SiC nanocomposites containing 5 vol% SiC. The mechanical properties were evaluated after SPS consolidation at 1450 °C and 40 MPa. A maximum bending strength of 980 MPa was achieved, compared to that of monolithic alumina of 350 MPa A hardness of 19 GPa was achieved at 1400 °C, which was the maximum hardness evaluated. There was no significance difference in the hardness values evaluated between 1400 and 1500 °C. Again, the fracture toughness was found to be improved (4.5 MPam^1/2^), while the monolithic alumina fracture toughness was 3.5 MPam^1/2^. Gao et al. explained that the overall improvement in mechanical properties can be attributed to the effective distribution of SiC nanoparticles in the alumina matrix and at the boundaries of the alumina. SiC particles at the boundary inhibit grain boundary movement and decrease the grain size of alumina, producing a fine microstructure and better mechanical properties. The most important mechanical properties typically evaluated are hardness and fracture toughness The Vickers hardness is typically used to determine the hardness of ceramics using the following equation [133]:(5)HV=1.854Pd2
where *P* is the applied load in N and *d* is the length of the diagonal of the indentation in mm.

From the hardness test result, the fracture toughness can be calculated using the crack length and indent length observed from the microscope. Antis’ equation below is used to evaluate the fracture toughness of monolithic and nanocomposite ceramics.
(6)$$K_{\mathrm{ic}}=0.016\sqrt{\frac{E}{H_{V}}}\frac{P}{C^{\frac{3}{2}}}$$
where *E* is the elastic modulus, *H*v is the Vickers hardness, *P* is the applied load, and *C* is the diagonal of the crack length [136].

Consolidation of 5 and 10 vol% SiC alumina–SiC nanocomposites was performed by Wang et al. [82]. After powder preparation by the precipitation method, the powders were hot pressed at 35 MPa at 1700 and 1750 °C for 5 and 10 vol% SiC, respectively. Mechanical properties, such as the strength, hardness, and fracture toughness, were evaluated. Maximum strength values of 467 and 415 MPa were achieved for the 5% and 10% SiC-alumina nanocomposite, respectively, while a strength of 350 MPa was achieved for monolithic alumina. The nanocomposite containing 5 vol% SiC exhibited the maximum fracture toughness (4.7 MPam^1/2^), while the 10 vol% alumina–SiC nanocomposites presented a lower fracture toughness (3.8 MPam^1/2^), which was an improvement compared to monolithic alumina (3.25 MPam^1/2^). Grain size refinements resulting from the positioning of SiC nanoparticles in the grains and at the grain boundaries of alumina were the reason for the improvement in the mechanical properties. Various fracture modes, such as a change in fracture mode, crack bridging and deflection and coefficient of elastic modulus and thermal mismatch between SiC and alumina, also contributed to the enhancement of the mechanical properties. A change in the fracture mode from intergranular to transgranular resulting from grain boundary strengthening was the reason for the improvement in fracture toughness, but as the volume fraction of SiC increased to beyond 5 vol% SiC, agglomeration occurred, resulting in the deterioration of the mechanical properties [46]. Densification results show that at a sintering temperature of 1700 °C, 5 vol% SiC-Al_2_O_3_ was fully densified, but the 10% SiC-Al_2_O_3_ nanocomposite was not fully densified until temperatures exceeded 1750 °C. The mechanical properties of Al_2_O_3_–SiC nanocomposites are presented in Table 1 while the hardness and fracture toughness variation with volume fractions of SiC and particle size is shown in Figure 2. In most of the reports, 5 vol% SiC addition to alumina exhibited the best mechanical performance, and beyond this composition agglomeration occurred, impairing the mechanical properties of the nanocomposites. Densification reduces with the addition of SiC to alumina, as SiC particles at grain boundaries inhibit grain boundary migration and grain boundary diffusion. This grain boundary inhibition effect becomes more pronounced with increasing SiC content.

The influence of SPS consolidation was studied by Oh et al. [90] after sintering at temperatures of 800–1500 °C, a pressure of 5.5 MPa and a heating rate of 30 °C/min. Hot pressing was also executed under the same sintering conditions. Higher densification was achieved via SPS than through hot pressing. The strength of the composite sintered by SPS reached 380 MPa, while monolithic alumina sintered by SPS and HP reached values of 200 and 160 MPa, respectively. Recently, Saheb and Mohammad [133] reported significant improvements in the hardness of alumina containing 5 wt% SiC after SPS consolidation at 1500 °C. The maximum hardness (21.78 GPa) was evaluated, while a lower value of fracture toughness (2.65 MPam^1/2^) less than that of monolithic alumina (3.61 MPam^1/2^) was reported. However, when the SiC concentration increased to 10 wt%, the fracture toughness increased to 3.64 MPam^1/2^. The presence of the SiC hard phase and microstructure refinement with SiC addition were proposed as mechanisms that increased the hardness. It is crucial to determine whether the method of powder preparation and the consolidation techniques employed contributed to the final mechanical properties of the nanocomposites. Again, it is pertinent to know the maximum mechanical hardness and fracture toughness reported by Dong et al. [137], as shown in Table 1. It is therefore highly recommended that for small concentrations of SiC, such as 5 and 10 wt%, in the alumina matrix, the hot pressing consolidation technique should be employed [127]. The SPS technique is suitable for high volume fractions of SiC in alumina, as the conductivity is improved. In all cases, homogenous distributions of second phase particles in the alumina matrix are essential to achieving optimal mechanical properties.

### 3.2. Mechanical Properties of Alumina–CNT Nanocomposites

It has been established that the range of CNT content in CNT/Al_2_O_3_ nanocomposites varies over a wide range from 0.01 to 35 wt%. Yamamto et al. [140], Sarkar and Das [110], Cha et al. [128] and Zhang et al. [111] found that MWCNTs start to form agglomerates at contents of over 0.9, 0.6, 0.18 and 1 vol%, respectively. The agglomeration of nanotubes at contents ranging from 2 to 7.39 wt% MWCNTs has been reported in the literature [43,140,141,142,143]. The concentration of SWCNTs that has better morphological precision and properties and a high clustering tendency is expected to be lower than the MWCNT concentrations in an alumina matrix that achieve the same effects [144].

The mechanical hardness of Al_2_O_3_/CNT nanocomposites varies significantly based on numerous factors that include the CNT content and dispersion extent, CNT purity, sintering temperature, extent of densification, nature of the interface formed, matrix grain sizes and applied indentation load [145]. The mechanical properties of alumina–CNT nanocomposites are presented in Table 2. Zhang et al. [111] investigated the mechanical properties of Al_2_O_3_/MWCNT nanocomposites consolidated by SPS at 1150 and 1450 °C. Optimal nanocomposite properties were achieved at 7.39 wt% MWCNTs. For instance, a hardness of 9.98 GPa was achieved for nanocomposites sintered at 1450 °C, while the hardness of pure monolithic alumina was measured at 9.21 GPa under the same conditions as shown in Figure 3. A maximum fracture toughness of 4.7 MPam^1/2^ was achieved for a nanocomposite containing 7.39 wt% MWCNTs sintered at 1450 °C. The mechanical properties of the monolithic alumina and nanocomposites were found to be lower at a low sintering temperature (1150 °C). Additionally, the densifications of the nanocomposites were all lower than that of the monolithic alumina sintered at the same conditions. Relative densities of 98.2% and 79.1% densification were achieved for monolithic alumina and 7.39 wt% MWCNT/Al_2_O_3_ nanocomposites, respectively. This increase in the mechanical properties of alumina with MWCNT addition can be attributed to effective dispersion and to the superior consolidation techniques employed. The reduction in mechanical properties can be attributed to the agglomeration of MWCNTs, which housed agglomerated pores that resisted densification. The tribological characteristics of hot pressed alumina–CNT nanocomposites have also been studied [43]. Hardness was found to increase with increasing CNT content, and maximum hardness was reached when the CNT content was 4 vol%. Poor cohesion and inhomogeneous distribution of CNTs spurred a reduction in hardness beyond 4 vol%. Wear loss became more pronounced after addition of 10 vol% CNTs, and the frictional coefficient decreased progressively as the CNT content increased from 0 to 10% CNT. This occurred because as the CNT content increases, the alumina grain size decreases, and most CNTs are located along the grain boundaries of alumina. Thus, a loss of cohesion between alumina is responsible for the deterioration of the mechanical properties. Several other researchers have found improvements in the mechanical properties of alumina reinforced with CNTs. Ahmad et al. [129,146] sintered 2 wt% MWCNT-alumina nanocomposites and reported a maximum hardness of 18 GPa for the nanocomposites, while a hardness of 16 GPa was evaluated for monolithic alumina. The fracture toughness was measured as 6.8 MPam^1/2^, but as the MWCNT content increased to 10 vol%, the fracture toughness decreased to 4.5 MPam^1/2^, while the monolithic alumina fracture toughness reached 3.2 MPam^1/2^. Sun et al. [21] also studied alumina-MWCNT nanocomposites containing various amounts of MWCNTs. The maximum fracture toughness (4.9 MPam^1/2^) was reported for the composite containing 0.1 wt% MWCNTs, while monolithic alumina presented a value of 3.7 MPam^1/2^. An investigation of the effects of the CNT distribution in the alumina matrix was reported by Ghobadi et al. [147] after pressureless sintering of alumina containing 3 vol% MWCNT. The mechanical properties of alumina were improved, and a maximum fracture toughness (4.7 MPam^1/2^) was achieved, while a fracture strength of 363 MPa was found. This improvement in the mechanical properties is attributed to crack bridging and nanotube pull-out mechanisms. Yamamoto et al. [148] also studied the influence of microstructures on alumina containing various fractions of MWCNTs densified by pressureless sintering. The volume fraction of MWCNTs was varied from 0 to 2.5 vol%, and the maximum bending strength (742.6 MPa) and maximum fracture toughness (5.83 MPam^1/2^) were achieved at 0.9 vol% MWCNT loading. Further increases in the content of MWCNTs to beyond 0.9 vol% result in decreasing mechanical properties. Thus, the authors agree that increasing the MWCNT concentration results in agglomeration and that agglomerated MWCNTs in the alumina matrix reduce densification. Again, clusters of MWCNTs exhibit poor load-carrying capacities, and this effect is identical to that found for pores. The mechanical performance of alumina containing varying amounts of MWCNTs (0.15, 0.3, 0.6, 1.2 and 2.4 vol%) was investigated by Sarka and Das [149] after consolidation by CIP at 1700 °C. Nanocomposites of 0.3 vol% MWCNT exhibited the strongest mechanical properties (a hardness of 19.52 GPa and a fracture toughness of 4.83 MPam^1/2^). Increasing the MWCNT concentration to 1.2 vol% resulted in a reduction in the hardness and fracture toughness to 18.65 GPa and 2.71 MPam^1/2^, respectively. The authors attribute this reduction in mechanical properties with CNT loading to the decreasing densification and agglomeration of CNTs with increasing CNT content. The mechanical strength of Al_2_O_3_–CNT nanocomposites as reported by several researchers [21,110,129,140,147,148,150] ranges from 260 MPa to 750 MPa. The strength of all nanocomposites evaluated exceeds that of the corresponding monolithic alumina.

Generally, Al_2_O_3_/CNT nanocomposites exhibit improved fracture toughness compared to pure Al_2_O_3_ up to a certain amount of CNT loading at which little or no CNT agglomeration occurs. Almost all the authors agree that the primary toughness mechanisms are realized through crack deflection, crack bridging, and nanotube pull-out processes. However, the contributions of thermal residual stress, as reported by Chen et al. [4], have been found to be negligible. Again, the lower coefficient of friction obtained for Al_2_O_3_/CNT nanocomposites than for pure alumina can be attributed to the self-lubricating properties of CNT and to their rolling and sliding at contact points. As presented in the table, the maximum fracture toughness (9.7 MPam^1/2^) was achieved for a sample sintered by SPS. Thus, it is reasonable to conclude that for the consolidation of alumina–CNT nanopowders, SPS is highly suitable. Furthermore, pressureless sintering is not a good technique for alumina–CNT nanopowder sintering, as the process is applied at elevated temperatures that can destroy CNTs.

## 4. Microstructures of Alumina-Based Nanocomposites

### 4.1. Microstructures of Alumina-SiC Nanocomposites

Microstructural and morphological factors contribute immensely to the properties of nanocomposites. In the alumina–SiC nanocomposite system, SiC nanoparticles can be dispersed among grains, at grain boundaries or simultaneously among grains and at grain boundaries of alumina. Several authors [11,12,81,82,153] have acknowledged the pinning effects of SiC particles on grains of the alumina matrix, thereby leading to the formation of fine microstructures and strong grain boundaries in alumina. The tremendous enhancement of the mechanical performance of alumina is mainly ascribed to the refinement of its microstructure, while SiC inhibits the grain boundary mobility of alumina. Gao et al. [81] studied the microstructure of alumina–SiC using a transmission electron microscope (TEM) and a scanning electron microscope (SEM) and found that SiC particles were homogenously dispersed throughout the alumina matrix. Large proportions of SiC particles were observed within Al_2_O_3_ grains, while a small number of large particles were found at grain boundaries. Wang et al. [82] conducted studies on the effects of nanoscale SiC particles on the microstructures of alumina, and SEM results for the failure surface showed a change in the fracture mode from intergranular to transgranular resulting from SiC additions to the alumina. TEM micrographs revealed that SiC particles were located within Al_2_O_3_ grains, while only a few particles were found at boundaries or at the junctions of grains. A field emission scanning electron microscopic (FESEM) characterization of Al_2_O_3_-17SiC nanocomposites was performed by Borrel et al. [86]. In their study, an Al_2_O_3_–SiC nanocomposite was thermally etched at 1500 °C prior to FESEM to reveal grains and grain boundaries. SiC was found to be distributed more at grain boundaries (Figure 4a) at a sintering temperature of 1400 °C. However, as the sintering temperature increased to 1500 °C, alumina matrix grain growth occurred, consuming more SiC and leaving less available at the grain boundaries (Figure 4b). This microstructural change due to the SPS temperature had a significance effect on the mechanical and functional performance of the Al_2_O_3_–SiC nanocomposites. A high-resolution transmission electron microscope (HRTEM) was also used by Sciti et al. [94] to study the distribution of SiC in the alumina matrix. It was found that more than 90% of SiC was located at the grain boundaries of alumina (Figure 5a), while few particles of SiC were located within alumina grains. Again, small agglomerates of SiC were found within grains of alumina. Evidence of strain contrasts was found around the grains of alumina due to the coefficient of thermal expansion difference between alumina and SiC particles, while a clear interphase boundary between SiC and alumina was found (Figure 5b).

Johnson et al. [120] investigated the phase stability of Al_2_O_3_–SiC nanocomposites by X-ray diffraction (XRD) of a composite sintered at 1400–1700 °C, a heating rate of 250 °C/min and 100 °C for the initial and final heating over a 5 min holding period. XRD analysis (Figure 6) was performed for the Al_2_O_3_-30% SiC and Al_2_O_3_-50% SiC nanocomposites. The results reveal a silicon oxide-containing phase in the composite investigated, identified as mullite, an aluminum silicate phase with chemical formula Al_2_SiO_5_. It is evident that increasing the SiC amount in the nanocomposites increases the mullite content. Furthermore, Wang et al. [154] noted that the stability and grain size of alumina are affected by the sintering temperature. At temperatures between 800 °C and 1000 °C the grain size of alumina was less than 50 nm, and the XRD results confirm that θ-alumina and γ-alumina coexisted. Further increases in the sintering temperature to 1200 °C increased the grain size of alumina to microscale sizes, while α-alumina was formed from the two phases mentioned above. The authors conclude that the formation of nanosized alumina at elevated temperatures is only possible by using suitable inclusions that restrict grain boundary sliding.

### 4.2. Microstructure of Alumina–CNT Nanocomposites

The microstructure of Al_2_O_3_/CNT nanocomposite has a dominant effect on the overall performance of the nanocomposites. Techniques commonly used to study microstructures of alumina–CNT nanocomposites include scanning electron microscopy (SEM) and transmission electron microscopy (TEM) [155]. In some cases, X-ray diffraction is used to measure the impurity and stability of CNTs during sintering. An et al. [43] studied the microstructure of Al_2_O_3_/CNT nanocomposites through a transmission electron microscope (TEM). TEM micrograph results show that the nanotube is a hollow multi-walled tube and not a solid fiber. The crystallite size of alumina tends to decrease as the CNT content increases. CNTs were found at grain boundaries. The cohesion between CNTs and alumina seems to be poorer as the CNT content increases. This poor cohesive force between CNTs and the alumina matrix results from the lack of densification that occurs as the CNT content increases. A reduction in the grain size of the alumina matrix occurred due to the adhesion of CNTs to alumina grains. Kumari et al. [156] investigated the thermal performance of alumina–CNTs consolidated by SPS. The microstructures of nanocomposites were studied by field-emission scanning electron microscopy (FE–SEM) (Figure 7) and TEM (Figure 8). TEM and HRTEM results showed that the alumina maintained their original size after CNT growth. Micrograph results revealed that the CNTs survived and maintained their integrity under the quickly elevated temperature of SPS. As observed in the FE–SEM images (Figure 7), the CNTs are clean and curly, while the alumina nanoparticles preserve their size and do not increase in size after SPS sintering. It is also evident that there is contact between the CNTs and the alumina matrix sintered at elevated temperatures (1450 °C), and this helps maintain the electrical conductivity of the nanocomposite. In addition, Yamamoto et al. [148] investigated the relationships between the microstructure and properties of alumina containing various amounts of MWCNTs. They found that the amount of grain boundary MWCNTs increases with increasing MWCNT amount regardless of the types of MWCNTs involved. When the MWCNT content increased from 0.5 to 1.9 vol%, the ratio of grain boundary MWCNTs increased from approximately 50% to 85%. The increasing content of grain boundary MWCNTs changed the mode of fracture from intergranular to transgranular fracture, promoting the fracture toughness of the nanocomposites.

Lee et al. [49] presented TEM images showing MWCNTs buried in an alumina matrix. They also observed that the grain size of the matrix decreases with an increasing amount of MWCNTs. It is crucial to determine how the final microstructure of alumina–CNT nanocomposites depends on the volume fractions of the CNTs, the grain size of the alumina powder, method of powder preparation and consolidation techniques employed. The improper distribution of CNTs in alumina can cause inhomogeneous heat distribution and eventually result in non-uniform sintering and microstructure formation, especially during the SPS process [127]. Again, such uneven heat distributions are dominant during the consolidation of electrically insulating materials, such as alumina. The addition of electrically conducting materials, such as CNTs, can promote heat distribution and significantly improve the homogeneity in the microstructure, consequently resulting in a smaller grain size. Thus, SPS is more sensitive to the CNT distribution than HP, as the uneven distribution of CNTs can result in localized heating, which produces inhomogeneous grain growth. Finally, for proper control of the microstructure, the careful selection and optimization of SPS parameters (heating rate, sintering pressure, sintering temperature and dwell time) are critical.

## 5. Thermal Properties of Alumina-Based Nanocomposites

### 5.1. Thermal Properties of Alumina–SiC Nanocomposites

Alumina–SiC nanocomposites have attracted considerable attention as a result of their potential applications, including in cutting tools, refractories for glass, gas radiant burners and wear resistant parts [85]. The application of alumina–SiC nanocomponents to all these applications requires that the thermal conductivity be as high as possible to reduce the risk of thermal shock failure. It is well established that the heat transport properties of materials rely strongly on the purity conditions of the crystal lattice of grains. Therefore, careful control of impurities emanating from raw materials and processing conditions must be achieved to enhance the thermal conductivity of materials. The thermal properties of thermal diffusivity, specific heat capacity, coefficient of thermal expansion and thermal conductivity are important parameters used in many applications of alumina-based nanocomposites [157].

#### 5.1.1. Heat Capacity

The heat capacity of a given material denotes a material’s ability to absorb thermal energy. Thermal energy includes the kinetic energy of atomic motion and potential energy of the distortion of interatomic bonds. Heat capacity *C* is express by the following equation [158]:(7)$$C=\frac{\Delta Q}{\Delta T}=\frac{dQ}{dT}$$
where *Q* is the amount of heat absorbed and *T* is temperature. The unit of heat capacity is J/K. Heat capacity can be measured at a constant temperature or a constant volume [158].

#### 5.1.2. Thermal Diffusivity

Thermal diffusivity is a thermophysical characteristic of materials that defines the velocity of heat transmission by conduction through the material during a change of temperature. Thermal diffusivity is related to the thermal conductivity, specific heat capacity and density of the material. Among the methods for measuring thermal diffusivity, the flash method is the most effective, as it requires short measuring time, is non-destructive and generates accurate results that are reproducible. The flash method for thermal diffusivity measurement involves the uniform radiation of a small specimen over its front face with a short pulse of energy [159]. Thermal diffusivity is expressed as:(8)$$\alpha =\frac{\lambda }{{\rho C}_{p}}$$
where α is the thermal diffusivity, λ is the thermal conductivity, ρ is the density and $C_{p}$ is the heat capacity of the material.

#### 5.1.3. Thermal Conductivity

Thermal conductivity as a mechanism of heat transfer can be defined as a material’s ability to conduct or transfer heat. The first general statement relating to heat flow and temperature gradients was made by Fourier in 1822. According to Fourier, for a material subjected to steady state heat flow, the quantity of heat (*q*) is related to the temperature gradient ($\frac{dT}{dx})$ by the expression [160],
(9)$$q=-\lambda \frac{dT}{dx}$$
where λ is the proportionality called the thermal conductivity with a unit of W/mK [161].

The mechanism of heat conduction in dielectric solids can be modeled with the aid of the Debye kinetic equation [162],
(10)λ=1/3Cvl

In equation (10) above, *C* is the specific heat per unit volume, *v* is the mean sound velocity and *l* is the average free path of phonons. It is important to state that the average free path decreases with an increase in temperature. The product of the velocity and the mean free path of a phonon is referred to as thermal diffusivity *D* [162]:(11)D=vl

Thus, thermal conductivity is defined as [162]:(12)$${\lambda \;=\;1/3C}_{P}\rho D$$

The mechanism of heat transfer by conduction through solids requires thermal energy. Carriers of heat transfer by conduction include electrons, phonons and photons. For non-metallic materials, phonons serve as carriers of heat through the thermal vibrations of atoms. However, in metals, heat conduction occurs via electrons, but in alloys and semiconductors, both phonons and electrons contribute to their thermal conduction [162].

#### 5.1.4. Models for Predicting the Thermal Conductivity of Nanocomposites

Different models are available for predicting the thermal conductivity of composites materials. The simplest of these models is the Maxwell model, which uses the rule of mixture to evaluate the thermal conductivity of composites. According to the Maxwell model [163], the thermal conductivity of composites ($\lambda_{c})$ is related to the thermal conductivity of the dispersed phase ($\lambda_{f}$) and the matrix ($\lambda_{m}$),
(13)$$\lambda_{c}=\varphi \lambda_{f}\;+\;(1\;-\;\varphi {)\lambda }_{m}$$
where φ is the volume fraction of the reinforcement phase [163,164].

In this model, the magnitude of thermal conductivity depends on the orientation of the filler in the matrix. Maximum thermal conductivity is achieved when the filler is arranged parallel to the heat flow. The thermal conductivity is at its lowest when the filler is in series with the direction of heat flow. The Maxwell model only considers the parallel arrangement of the filler with the direction of heat flow, and as such, the model overestimates the thermal conductivity of a composite system.

The Cheng–Vachon model [165] developed for estimating the thermal conductivity of composites, where λ_f_ > λ_m_, is given by [165]:(14)$$\frac{1}{\lambda_{c}}=\frac{1-B}{\lambda_{m}}\;+\;\frac{1}{\sqrt{{K\lambda }_{d}\left(\lambda_{f}{\;+\;B\lambda }_{d}\right)}}\mathrm{In}\left[\frac{\sqrt{\lambda_{m}{\;+\;B\lambda }_{d}}\;+\;B/2\sqrt{{K\lambda }_{d}}}{\sqrt{\lambda_{m}{\;+\;B\lambda }_{d}}-B/2\sqrt{{K\lambda }_{d}}}\right]$$
where $B=\sqrt{3\varphi /2}$, $K=-4\sqrt{\frac{2}{3}\varphi }$ and $\lambda_{d}{\;=\;\lambda }_{f}{\;-\;\lambda }_{m}$.

The model has been determined to be more accurate, and thermal conductivity values evaluated using this model are in better agreement with experimental results. However, the model does not take into consideration the geometry and size of particles, and as such, it underestimates the thermal conductivity of composites. Geometry and size limitations of the Cheng–Vachon model [165] brought about the development of Hamilton and Crosser’s model [166]. This model takes into consideration the sphericity (ψ) of particles. Hamilton and Crosser’s model is given by [166]:(15)$$\lambda_{c}{\;=\;\lambda }_{m}\left(\frac{\lambda_{f}{\;-\;(n-\;1)\lambda }_{m}{\;+\;(n\;-\;1)(\lambda }_{f}{\;-\;\lambda }_{m})\phi }{\lambda_{f}{\;+\;(n\;-\;1)\lambda }_{m}{\;-\;(\lambda }_{f}{-\;\lambda }_{m})\phi }\right)$$
where n is a constant, n=3/ψ and for a spherical particle, n=3.

Lewis and Nielsen [167] developed a mechanical model that considers the particle size, particle shape and packing arrangement of particles in a matrix. According to Lewis and Nielsen’s model for composites [167],
(16)$$\lambda_{c}{\;=\;\lambda }_{m}\frac{1\;+\;\xi \eta \phi_{f}}{1\;-\;\Psi \eta \phi_{f}}$$
where $\eta =\frac{\lambda_{{f-\lambda }_{m}}}{\lambda_{f\;}{+\;\xi \lambda }_{m}}$ and $\Psi =1\;+\;\frac{(1-\emptyset_{m})}{\emptyset_{m}^{2}}\varphi_{f}$, ξ is the shape factor that depends on the shape, orientation and aspect ratio of second phase dispersed particles, $\emptyset_{m}$ is the packing fraction of the dispersed phase, and Ψ denotes the maximum concentration achievable in the second phase. For randomly packed spherical particles, ξ = 1.5 and Ψ = 0.637 [168].

The Lewis–Nielsen model is suitable for nanocomposite systems, as it considers thermal resistance. The effect of thermal resistance has been incorporated into a two-phase system by introducing Kapitza’s resistance *R*_k_ in series with the particle resistance, dλf , where d is the size of the particle. The resulting resistance is then given as [168]:(17)dλF′=dλf + Rk

Therefore, the effective thermal conductivity of a particle including the interfacial resistance is given as [168]:(18)λF′=λf1 + RKλfd

When a particle is very small, the term Rk d converges to ∞, and as such, the filler does not contribute to the thermal conductivity of the system, as the effective thermal conductivity of the particle is zero ($\lambda_{F}^{\prime }$ = 0). For large particles, the interfacial resistance is insignificant as Rkd→0.

This explains why most thermal conductivity models fail to predict the thermal conductivity of nanocomposite systems.

In alumina–SiC nanocomposites, the transport properties (heat and electrical) are determined by various critical factors, such as the volume fraction of SiC, impurity levels of other materials, parameters of the final microstructure, particle size of alumina matrix grains and distribution (intergranular, intragranular or both) of SiC inclusions. Previous reports on the thermal conductivity of Al_2_O_3_–SiC, listed in Table 3, show that the thermal conductivity increases with SiC content.

The thermal conductivity of SiC-containing compounds is difficult to model, as only ambient temperature thermal conductivity data are available for SiC materials in the literature. Depending on the impurity content and processing techniques applied, the thermal conductivity of SiC can vary from 40 to 100 W/mK [157,170]. Higher thermal conductivity values of 150–325 W/mK [29,171,172] have been achieved for the intrinsic thermal conductivity of SiC, evaluated from the heat conduction model in the presence of an interfacial thermal barrier.

Barea et al. [169] studied the thermal diffusivity of Al_2_O_3_–SiC nanocomposites containing 30 vol% SiC after hot pressing at 1550 °C for 60 mins at a sintering pressure of 50 MPa. The room temperature thermal diffusivity improves with SiC platelet volume fractions from 0.092 cm^2^/s for pure alumina to 0.153 cm^2^/s for the 30 vol% SiC-Al_2_O_3_ composite. Again, for every composite, the thermal diffusivity decreases with an increase in the measurement temperature, reaching values of 0.015 cm^2^/s and 0.028 cm^2^/s at 1000 °C for 0 and 30 vol% SiCpl, respectively. It was also confirmed that the thermal diffusivity measured in a perpendicular configuration is higher than the parallel one at all temperatures. In addition to the thermal diffusivity, the thermal conductivity, λ, was evaluated, and a maximum thermal conductivity value of 42 W/mK was achieved at room temperature for 30 vol% platelet content measured in the parallel direction, which is 52% higher than the value found for monolithic alumina (28–35 W/mK) [173]. A λ value of 49 W/mK was found for the perpendicular direction. Thus, the heat flow in the perpendicular direction is greater than that in the parallel direction. This occurs because anisotropy in the thermal conductivity is expected for hot pressed Al_2_O_3_–SiC, as the phonon mean free path should be greater in the basal plane that correlates with the platelet facet, promoting heat conduction in the perpendicular direction. The availability of grain boundary interfacial resistance can explain the increase in λ estimated for the 30 vol% sample measured in the perpendicular direction. The influence of thermal resistance on the Al_2_O_3_/SiCpl interface declines with temperature when thermal stresses are eliminated, and phonon-phonon scattering becomes a dominant factor shaping the phonon mean free path. This explains why the difference in thermal conductivity found for the parallel and perpendicular directions decreases with rising temperature. In addition, the decline in the thermal conductivity with temperature is attributed to an increase in crystal lattice vibrations with temperature, increasing the scattering of phonons. Thus, Barea and colleagues concluded that thermal conductivity is independent of the shape and size of the dispersing phase.

The variation of thermal conductivity with the SiC content has also been investigated by Fabbri et al. [157] by hot pressing Al_2_O_3_–SiC containing 28–39 vol% SiC whiskers at 1900 °C and 50 MPa for 20 to 60 mins of sintering time. Room temperature thermal conductivity measurements were performed (29 W/mK) for monolithic alumina hot pressed at 1500 °C, while the values obtained for composites were 40 and 30 W/mK for two different SiC contents. The thermal conductivity was found to improve with SiC content and purity. The thermal behavior of Al_2_O_3_–SiC nanocomposites containing 3–20 vol% SiC was studied by Parchoviansky et al. [85]. In their work, alumina–SiC powders were hot pressed at 1740 °C and a sintering pressure of 30 MPa, while alumina monolithic was sintered at 1350 °C. The thermal diffusivity increased with the volume of reinforcement phase particles. At ambient temperature, the thermal diffusivities of Al_2_O_3_ and 20 vol% SiC/Al_2_O_3_ were 0.093 cm^2^/s and 0.135 cm^2^/s, respectively, regardless of the size and geometry of SiC inclusions. The thermal diffusivity and thermal conductivity were evaluated as a function of temperature, as shown in Figure 9 and Figure 10, respectively. The highest value of λ at room temperature was obtained for nanocomposites AS20c and AS20f (38 W/mK), while the alumina reference thermal conductivity value was only 28 W/mK. Comparing the theoretical thermal conductivity (700 W/Mk) evaluated for SiC with those for single crystal (490 W/mK) [174] and polycrystalline SiC evaluated for a hot pressed sample (270 W/mK) [175], the resulting thermal conductivity of a nanocomposite containing 20 vol% SiC should be much higher. This is because the main factor affecting the thermal conductivity is phonon scattering, with SiC additions thus appearing as scattering sites. Phonons are also related to defects such as lattice defects, grain boundaries, and other microstructural defects. The addition of second phase particles, a higher level of residual porosity resulting from low densification of the nanocomposites, and the change in the nature of the interface negatively affect the thermal conductivity. The existence of thermal mismatch between SiC and alumina and a thermal barrier at the interface between the alumina matrix and the SiC interface also impairs the thermal conductivity. In conclusion, the thermal conductivity of alumina–SiC nanocomposite systems is mainly affected by the volume content of SiC and not by the average size of SiC particles included.

### 5.2. Thermal Properties of Alumina–CNT Nanocomposites

Carbon nanotubes have been identified by researchers as potential fillers in the reinforcement of ceramic matrices due to their intrinsically high thermal conductivity. The thermal conductivity values of SWCNTs and MWCNTs are 6000 and 3000 W/mK, respectively [18,176]. However, the addition of CNTs to alumina does not generate as high a thermal conductivity as expected based on the intrinsic thermal conductivity of CNTs. There is no general trend regarding improvements in the thermal conductivity of Al_2_O_3_/CNT nanocomposites. A substantive improvement in the thermal conductivity (90.44 W/mK) of Al_2_O_3_/CNT nanocomposites was reported by Kumari et al. [112] for 7.39 wt% MWCNT/Al_2_O_3_ nanocomposites over monolithic alumina after SPS consolidation at 1550 °C. The further loading of MWCNTs to 19.1 wt% resulted in a decrease in the thermal conductivity (36.77 W/mK) of Al_2_O_3_/MWCNT nanocomposites even at higher sintering temperatures. Furthermore, Zhang et al. [111] studied the thermal conductivity of Al_2_O_3_/SWCNT nanocomposites densified by spark plasma sintering with CNT content ranging from 5 to 15 vol%. The thermal diffusivity and conductivity were investigated, and a thermal conductivity of 11.4 W/mK was achieved for 10% SWCNT; this value decreased to 7.3 W/mK as the CNT content increased to 15%. The thermal conductivity of pure alumina was evaluated as 27.3 W/mK. The decline in thermal conductivity was attributed to differences in the thermal characteristics of individual tubes and ropes. The effects of temperature on the thermal diffusivity (Figure 11a) and thermal conductivity (Figure 11b) have been investigated by Kumari et al. [112], who found that both thermal diffusivity and thermal conductivity decrease with measurement temperature, as shown in Figure 11. This decrease in thermal conductivity with measurement temperature occurs due to the increased scattering of phonons resulting from an increase in crystal vibration with the increase in temperature. In another report, Sarkar and Das [149] studied the influence of CNT loading and sintering temperature on the thermal conductivity of alumina. The thermal conductivity of pure alumina sintered at 1500 °C by CIP reached 29.39 W/mK, and it increased to 38.63 W/mK as the sintering temperature was increased to 1700 °C. Again, the addition of 0.15 vol% MWCNTs to alumina sintered at 1700 °C increased the thermal conductivity to 47.14 W/mK (22%). Further increases in the MWCNT content to 2.4 vol% decreased the thermal conductivity value to 5.05 W/mK (−86.93%). The authors attributed this reduction in thermal conductivity with CNT addition to the scattering of phonons by residual pores, the low thermal conductivity of highly clustered CNTs and the presence of kinks or twists in agglomerated CNTs.

Interfacial thermal resistance is one factor that affects the thermal conductivity of CNT/Al_2_O_3_ nanocomposites. A small value of interfacial resistance can significantly reduce heat transport through CNT/Al_2_O_3_ nanocomposites. The reported interfacial thermal resistance across the alumina–CNT matrix is approximately 8.3 × 10^−8^ m^2^ K/W [177]. This result implies that SWCNTs (6000 W/mK) cannot induce a greater thermal conductivity improvement in SWCNT nanocomposites compared to MWCNT nanocomposites. In addition, tube-tube interactions reduce the phonon mean free path of SWCNTs (−0.5–1.5 µm), and as such, the thermal conductivity value for SWCNTs is lower [178]. However, the phonon mean free path (20–500 nm) estimated for MWCNTs is much smaller than that for SWCNTs; thus, the phonon mean free path of MWCNTs in the alumina matrix is not affected by tube–tube interactions [134,179,180]. The main factors responsible for the marginal enhancement of the thermal conductivity of Al_2_O_3_-CNT nanocomposites include the scattering of phonons by residual pores, the tube–tube interactions of CNTs, the blocking of phonons by kinks or twists in agglomerated CNTs, the scattering of phonons by the interfacial resistance and high-temperature scattering due to increasing crystal vibrations. Sivakumar et al. [181] reported that the major factors responsible for the decrease in CNT-ceramic-based nanocomposites include the high interface thermal resistance between the nanotubes and ceramic matrix and between nanotubes. Table 4 shows the thermal conductivities of alumina–CNT nanocomposites, including the densification and consolidation types. It is clear from the table that not much work has been done on the thermal conductivity of alumina–CNT nanocomposites to the best of the authors’ knowledge. Recently, Saheb and Hayat [135] reported a decrease in the thermal properties of alumina with the addition of SiC and MWCNTs consolidated by SPS at 1500 °C. The thermal conductivity of monolithic alumina was reported as 34.44 W/mK, which reduced to 21.2 W/mK and 20.4 W/mK, respectively, when 1 wt% and 2 wt% MWCNTs were added to alumina containing 5 wt% SiC. The authors attributed this reduction in thermal conductivity to densification and microstructural refinement. Shin et al. [176] also studied the influence of SWCNTs on the thermal conductivity of alumina and noted that the thermal conductivity of alumina decreased with SWCNT inclusion. This decrease in alumina with SWCNT addition was attributed to a decrease in sound speed because of the defect in SWCNTs during SPS processing that reduced the phonon mean free path. This in turn reduced the thermal conductivity of nanocomposites.

In conclusion, the thermal properties of alumina-based nanocomposites depend strongly on the powder processing methods and consolidation techniques employed. However, an issue related to powder processing concerns the agglomeration of nanopowders during drying, and this could be ameliorated by employing freeze drying. Furthermore, SPS consolidation of alumina–SiC nanopowders produces nanocomposite of refined microstructures, and this reduces the thermal properties. To produce alumina–SiC nanocomposites with improved thermal properties, the use of HP is highly recommended for consolidation. Finally, to produce alumina–CNT nanocomposites with improved thermal properties prepared by conventional powder mixing, the concentration of CNTs should range between 0.3 and 0.9 wt% to avoid agglomerations, which can impair the thermal conductivity.

## 6. Electrical Properties of Alumina-Based Nanocomposites

### 6.1. Electrical Properties of Al_2_O_3_–SiC Nanocomposites

The Al_2_O_3_–SiC nano/microcomposites are composed of an Al_2_O_3_ dielectric as a matrix and a semi-conductive SiC reinforcement phase responsible for electrical conduction. It is expected that the electrical conductivity of a nanocomposite material system will depend on the intrinsic electrical conductivity of the component phases. The inclusion of conductive or semi-conductive reinforcement phases in a non-conductive (dielectric) system shapes the electrical conductivity of a nanocomposite system [182,183]. Percolation theory [184] explains correlations between conductive and volume fractions of randomly dispersed conductive or semi-conductive particles in a nonconductive matrix. Percolation theory involves the estimation of the charge formation of a conductive path network consisting, in this case, of equally interconnected SiC particles. The mechanism of electrical conduction in Al_2_O_3_–SiC nanocomposite materials lies in the establishment of a continuous network of SiC nanoparticles within the Al_2_O_3_ matrix. It is well known that one of the major parameters for achieving electro-conductive materials is the uniform distribution of SiC nanoparticles [185]. Furthermore, the position of the SiC nanophase is affected by alumina matrix growth. An increase in alumina grain size from powder to dense material can favor an increase in the relative volume fraction of SiC nanoparticles in the intragranular position. McLachlan [184] developed a mathematical model and estimated the minimum volume content of SiC creating a conducting path to be 17 vol%. Sawaguchi et al. [186] studied the electrical properties of Al_2_O_3_/SiC nanocomposites and achieved a resistivity of 10^13^ Ώcm for nanocomposites with up to 10 vol% SiC particles. However, when the volume content of SiC particles increased to 20 vol%, the electrical resistivity reduced to 10^6^ Ώcm. Parchoviansky et al. [85] investigated the influence of the volume fraction of SiC particles (Figure 11) on the electrical conductivity of Al_2_O_3_–SiC nanocomposites. The maximum electrical conductivity (4.05 × 10^−2^ S/m) was achieved at 20 vol% SiC, while the electrical conductivity of pure Al_2_O_3_ reached only 7.80 × 10^−6^ S/m. Lux [185] noted that in Al_2_O_3_/SiC nanocomposite systems, electricity is conducted through a complete network of SiC particles located inside Al_2_O_3_ matrix grains or at grain boundaries. This observation is also in line with the results of Parchoviansky et al. [85]. Figure 12 shows that at low volume fractions (<10 vol%) of SiC particles (both coarse and fine grained), the electrical conductivity improved slightly with SiC volume. As shown in Figure 12, at low SiC content, SiC particles did not interlinked to form a complete conductive path network. As the SiC content increased, the electrical conductivity was recorded, and a percolation threshold was achieved between 5 and 10 vol%. Parchoviansky et al. [28] concluded that the electrical conductivity is shaped by the grain size of the Al_2_O_3_ matrix but is independent of the grain size of SiC particles. Table 5 shows the electrical conductivity, densification, and consolidation techniques of Al_2_O_3_–SiC nanocomposites.

The effects of sintering temperature on the microstructures and electrical conductivity of alumina-17 vol% SiC nanocomposites consolidated by SPS have been investigated by Borrell et al. [86]. When sintering was applied at 1400 °C, a 99.1% relative density was achieved for a 25.7% SiC volume fraction placed in the intragranular positions. An increase in sintering temperature up to 1550 °C enhanced grain growth (from 430 to 590 nm); however, the SiC volume fraction at the intragranular positions remained unchanged (26.2%). A decrease in the effective SiC content at grain boundaries from 17 to 12.6 vol% due to the growth of alumina grains resulted in 25% of the SiC content changing from intergranular to intragranular positions, which promoted grain growth as shown in Figure 13a,b. It is important to note that the increase in the size of SiC particles during the SPS process is negligible, and the possibility of electrical tunneling is related to the probability of forming a complete network of SiC particles within the Al_2_O_3_ matrix. Two factors influence the formation of a complete path network. First, the alumina grain size provides the available grain boundaries to be taken over by the SiC nanophase, and second, the volume fraction of SiC particles in grains of alumina limits the silicon carbide nanoparticles forming the complete conducting path network. The nanocomposite consolidated at 1400 °C has an electrical resistivity of 31 Ώm (<100 Ώm), while that sintered at 1500 °C presents a value of 170 Ώm (>100 Ώm), and this difference is attributed to their different microstructures. In conclusion, SPS consolidation techniques can be used for microstructural design, and increasing the content of SiC in alumina enhances the electrical conductivity of nanocomposites.

### 6.2. Electrical Conductivity of Alumina–CNT Nanocomposites

Researchers have attempted to improve the electrical conductivity of purely monolithic alumina by incorporating highly conductive CNT into alumina. A dramatic increase in the electrical conductivity of Al_2_O_3_/CNT nanocomposites over that of pure Al_2_O_3_ was achieved when the CNT loading of the matrix reached a percolation threshold. The percolation threshold is the minimum amount of conductive phase necessary to form a connected network within a ceramic matrix [22,114,187]. The lowest percolation threshold of 0.64 vol% was reported by Rule et al. [22] after hot pressing CNT-MgAl_2_O_4_, while values of 0.18 wt% for Al_2_O_3_ [49] and of 1.7 wt% for ZrO_2_ [23] were found for CNT nanocomposites. This percolation threshold is roughly 20 times lower than that of microscale composites, and this small percolation threshold is considered to be a result of the large aspect ratio of CNTs.

Increasing demand for ceramics with high electrical conductivity has spurred researchers and scientists to devise means of tailoring the electrical conductivity of alumina–CNT nanocomposites. The high electrical conductivity of SWCNTs (>10^6^ S/m) and MWCNTs (>10^5^ S/m) [16,19] has had a tremendous effect in changing the electrical behaviors of ceramics from insulators to semiconductors. However, to explore the intrinsic electrical conductivity of CNTs in ceramic-CNT nanocomposites, the homogenous dispersion of CNTs in matrices is crucial. Homogenous distributions of CNTs in ceramic matrices result in the formation of continuous networks around ceramic matrix grains. Again, the available continuous network of the conducting path of CNTs is shaped by the microstructure (the grain size) of the matrix [25]. High electrical conductivity (576 S/m) has been reported for the alumina–CNT nanocomposite by Inam et al. [114] after spark plasma sintering of alumina containing 5 wt% MWCNTs. Kumari et al. [156] consolidated MWCNT/Al_2_O_3_ nanocomposites by applying SPS techniques at a sintering temperature of 1150 and 1450 °C, and they found that the electrical conductivity increases with an increase in the MWCNT content. The maximum conductivity of alumina–CNT nanocomposites at room temperature was measured as 3336 S/m for 19.1% MWalumina–CNT nanocomposites sintered at 1450 °C. The electrical conductivity also increases with sintering temperature. For instance, 7.39 wt% MWCNTs sintered at 1150 and 1450 °C presented electrical conductivity values of 288 and 705 S/m, respectively. Zhan and Mukherjee [115] investigated the electrical conductivity of alumina reinforced with single-walled carbon nanotubes containing 5–15 vol% SWCNTs. The electrical conductivity of Al_2_O_3_/SWCNT nanocomposites was found to increase with the SWCNT content. An electrical conductivity value of 1050 S/m was achieved for 5.7 vol% SWCNTs at room temperature. Again, the electrical conductivity was found to increase to 1510 S/m as the SWCNT content increased to 10% and to 3345 S/m as the SWCNT content was further increased to 15%, while pure alumina presented an electrical conductivity value of 10^−10^–10^−12^ S/m [115]. The dramatic improvement in the electrical conductivity of alumina reinforced with CNT was investigated by Ahmad and Pan [188]. They reported an increase in electrical conductivity with increases in the MWCNT content. Ahmad and Pan explained that increases in the multi-walled CNT content enhance the number of interconnected connections in the network, leading to the formation of several conducting paths. Furthermore, the electrical conductivity of nanocomposites was found to increase with measurement temperatures, as carrier mobility and thermally generated carrier formation increase as the temperature rises. The electrical conductivity, consolidation techniques and densification of alumina–CNT nanocomposites as reported in various papers are illustrated in Table 6. Lee et al. [49] investigated the electrical conductivity of Al_2_O_3_/MWCNT nanocomposites obtained by hot pressing. According to their report, the addition of 0.18 wt% (0.55 vol%) MWCNTs to the alumina matrix results in a sudden increase of the DC electrical conductivity by almost 12 orders of magnitude, which is the percolation threshold. Again, at between 1.0 and 2.0 wt% MWCNTs, no momentous change in electrical conductivity was observed. Furthermore, Sarkar and Das [149,189] recently reported an increase in the DC electrical conductivity of alumina with the addition of MWCNTs densified by CIP. A maximum electrical conductivity (0.1 S/m) was achieved with 2.4 vol% MWCNTs densified at 1700 °C. The authors noted that a percolation threshold was reached between 0.6 and 1.2 vol% CNT. Additionally, a lower percolation threshold was achieved at a higher sintering temperature because at lower sintering temperatures, the specimens shrank less, and CNT interconnections were difficult to achieve.

In Al_2_O_3_/CNT nanocomposites, electrical conduction is typically controlled by fluctuation-induced tunneling and the variable range hopping mechanism through interconnected CNTs at grain boundaries. The positioning of CNTs in the ceramic matrix is very important (i.e., whether CNTs are located at the grain boundary or in the grains of the matrix). For the former case, CNTs at the grain boundary establish an effective interconnected conductive network to convey electrons even at very low contents, and this promotes electrical conductivity [40,184,190]. For the latter case, CNTs do not form interconnected paths to establish conductive channels, and as such, percolation is not achieved, leading to a reduction in electrical conductivity. The percolation thresholds of MWCNTs in an α-alumina matrix usually range from 0.18 to 0.25 wt% MWCNTs [191]. However, in nanocomposites with spherical conductive fillers, such as SiC, the percolation threshold is approximately 5 wt% [114]. Thus, the minor percolation threshold value found for MWCNTs can be attributed to their high aspect ratio. In addition, most MWCNTs are located at grain boundaries, and this allows a small number of MWCNT to form a conducting path network for the electrons. The microstructure of the matrix plays a significant role in the electrical conductivity of the nanocomposite [192]. As reported by Inam et al. [114], the electrical conductivity of an alumina–CNT nanocomposite was found to increase by increasing the crystallite size of alumina while the CNT content was fixed, and the density of CNTs at the grain boundaries increased. The electrical conductivity was also improved by increasing the amount of CNTs while keeping the grain size constant. The above points show that the electrical conductivity of an alumina–CNT nanocomposite is dependent on the CNT concentration at the grain boundary, which serves as a conducting path. In conclusion, the electrical conductivity of Al_2_O_3_/CNT nanocomposites is affected by the composition of the CNT (CNT content), the microstructure of the matrix (grain size and grain boundary) and the sintering temperature.

## 7. Applications of Alumina-Based Nanocomposites

The various industrial applications of ceramics with appreciable electrical conductivity include static charge dissipation, ceramic heating, electric discharge machining, semiconductor wafers, electromagnetic radiation, aerospace components, electrode fuel cells, vacuum induction crucibles and structural applications [114,188]. In static charge dissipation, alumina–SiC and alumina–CNT nanocomposites with appreciable electrical conductivity showed exceptional performance as compared to monolithic alumina. Moving air is mostly saturated with static charges and this get accumulated when it comes in contact with an object. For an insulating material, such as pure alumina, the dissipation of such excessive charge is difficult leading to charge accumulation and eventually spiking. However, materials with appreciable electrical conductivity such as alumina–SiC and alumina–CNT, can dissipate charge faster and this avoid the problem of charge accumulation. Furthermore, ceramics materials are generally hard to machine and to manufacture complex-shape ceramic products, a material with high level of electrical properties is required. Electro discharge machining (EDM) of ceramic articles to near net shape is possible with the development of nanocomposites material of intrinsic electrical properties in addition to its superior mechanical performance. EDM mean of fabrication ceramic material is possible if the electrical conductivity is between 0.3 and 1 S/m. Alumina–CNT nanocomposites, due to their high electrical conductivity relative to that of alumina–SiC nanocomposites, are suitable for electro-discharge machining manufacturing.

Alumina–SiC nanocomposites with appreciable mechanical properties (hardness and fracture toughness) and thermal conductivity are suitable for the fabrication of cutting tool materials for machining [85,95], as such materials are highly resistant to fracture and wear. Improvements in the thermal properties of alumina with SiC addition can prevent the developed cutting tools from experiencing thermal shock-related failure. Additionally, due to the chemical stability of alumina–SiC nanocomposites, this method can be equally used for the development of biomedical implants. Optical sensors have been produced from carbon nanotube-alumina nanocomposites by Sehrawat et al. [193,194]. The principle of sensor sensitivity measurement is based on interaction of light with the crystal structure of material. Light illumination on the sensor material generates the following effects: (i) the photons with appreciable energy are absorbed within the sample which produce abundant free charge carriers (electrons and holes); (ii) photo-produced charge carriers are moved towards the electrodes when suitable bias is applied to the material; and (iii) finally the resistance of the material decreases and the conductivity is enhanced. The more the photon absorption, the greater the sensitivity of the material. The sensitivity of sensors increased with the CNT concentration, and the maximum sensitivity (13.2%) was achieved at 1.5 wt% CNT. However, increasing the contents CNT beyond 1.5 wt% result in the CNT agglomeration owing to their strong Van der Waals attraction among them. As a result of this agglomeration, only few portions of the sample interacted with the light and eventually few charge carriers are generated resulting in reduce sensitivity. Thus, alumina–CNT nanocomposites sensors can be fabricated through careful control of CNT concentration and proper dispersion to avoid agglomerations. Highly sensitive optical sensors developed from alumina–CNT nanostructured materials are widely used in the chemical, medical, biomedical, health care and imaging fields.

## 8. Conclusions

The research progress on the processing and characteristics of alumina–CNT/SiC nanocomposites has been reviewed. Central to the improvement of mechanical and functional properties is the uniform distribution of SiC and CNT in the alumina matrix. Uniform dispersion of SiC and CNT nanophases at low concentrations in the alumina matrix can be achieved with the aid of an ultrasonicator and planetary ball milling. However, at higher concentrations, the nanophase (SiC and CNTs) tends to agglomerate, and colloidal processing is the most effective tool for obtaining a uniform distribution of the nanophase in the alumina matrix. The combination of gum arabic and sodium dodecyl sulphate solutions has proven effective for the uniform dispersion of CNTs in the alumina matrix due to their electrostatic and steric repulsion effects, which lead to the de-agglomeration of CNTs. Furthermore, the consolidation of alumina–SiC and alumina–CNT nanopowders into bulk nanocomposites remains a serious challenge. Conventional techniques, such as pressureless sintering, are highly deleterious for the sintering of alumina-based nanocomposites, as the presence of excess heat tends to promote grain growth and eventually result in the loss of nanofeatures and of interesting properties of nanocomposites. For alumina–SiC nanopowders, consolidation hot pressing is highly recommended, as heating is applied by conduction, radiation and convectional means of heat transfer that promote a uniform heat distribution. In the case of highly conductive nanophase CNTs in an alumina matrix, SPS processes are the most suitable, as the conductivity of this material plays a significant role in its overall heating. Microwave sintering is not a suitable technique for alumna-based nanopowder sintering due to the dielectric properties of alumina. Several mechanisms have been deemed responsible for the enhanced mechanical performance of alumina–SiC nanocomposites, including crack deflection, crack bridging, changes in failure modes, thermal mismatch, microstructural refinement, Zener effects, etc. Significant improvements to the mechanical properties (23% increment in hardness and 129% increment in fracture toughness) of alumina containing 5 wt% SiC densified by hot pressing have been reported. Similarly, 12% and 32% increments in the mechanical hardness and fracture toughness, respectively, of alumina containing 2 wt% MWCNT after SPS consolidation have been reported. The thermal and electrical properties of alumina–SiC and alumina–CNTs are strongly dependent on the uniform distribution of SiC and CNTs in the alumina matrix. The enhanced thermal and electrical properties of the nanocomposite are attributed to the formation of conductive network paths of SiC through the grain boundaries of alumina. Once the percolation threshold is reached, a change in the electrical behavior of alumina from dielectric to semiconducting occurs. For alumina–SiC nanocomposites, the percolation threshold is reached when the concentration of the SiC nanophase is between 5 wt% and 10 wt%. However, for alumina–CNT nanocomposites, the percolation threshold exists within the CNT concentration range of 0.6–0.9 wt%. Functional properties (electrical and thermal) have been found to rely on nanocomposite processing, the microstructure, the composition of the reinforcement phase, the porosity, impurities, the measurement temperature, and the interfacial resistance between SiC and alumina. The thermal conductivity of alumina–CNT nanocomposites must be researched further, as the optimal CNT content in alumina that enhances thermal conductivity has not been defined. The highest thermal conductivity reported thus far is 90.44 W/mK for a CNT content of 7.39 wt%, and this decreases as CNT content increases to 19.1 wt%. However, the electrical conductivity of alumina–CNT nanocomposites increases with increasing CNT content, and this trend is clearly reported in the relevant literature.

CNTs have a significant effect on both the mechanical and thermal properties of alumina because of their high aspect ratios. A small amount of CNT has been found to enhance the mechanical, electrical and thermal properties of alumina. CNTs control the microstructure of alumina through grain refinement, as CNTs have a pinning effect on the alumina matrix. This effect becomes more pronounced as the CNT content increases in the nanocomposites. To achieve improved mechanical and functional performance in alumina–SiC and alumina–CNT nanocomposites, the following should be taken into consideration:(1)SiC and CNT must be effectively distributed in alumina to avoid agglomeration.(2)A combination of techniques, such as sonication and ball milling, is effective in the homogenous distribution of SiC in alumina matrices, especially at low nanophase concentrations.(3)Ultra-sonication, ball milling and molecular level mixing are suitable for dispersing CNTs in ceramic matrices at low concentrations of CNTs. However, for higher volume fractions of CNTs in alumina matrices, colloidal heterocoagulation and flocculation are highly recommended as means to facilitate homogenous distribution.(4)SiC and CNT concentrations must be optimized to enhance the mechanical and functional properties.(5)Appropriate consolidation techniques must be employed to prevent grain growth and the formation of inhomogeneous microstructures that lead to poor mechanical and functional performance.

To produce nanocomposites with high electrical conductivity, the concentration of the second nanophase (SiC or CNT) should exceed the percolation threshold.

## Figures and Tables

**Figure 1 nanomaterials-09-00086-f001:**
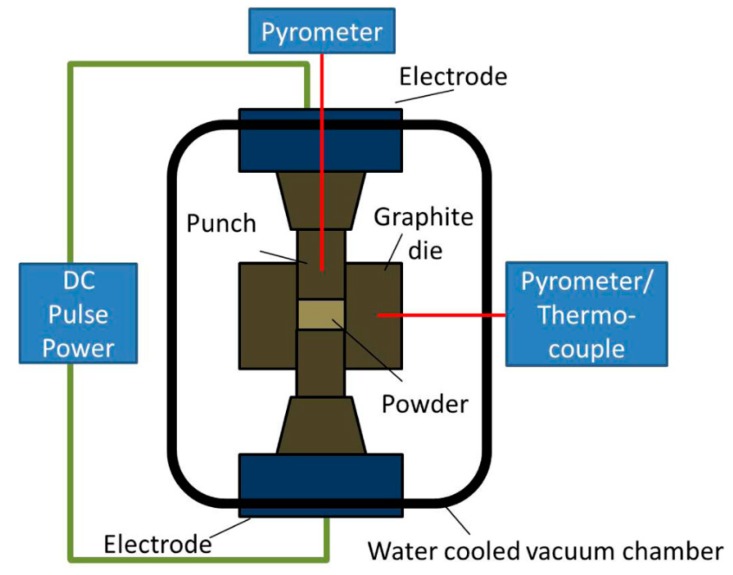
Principles and mechanisms of the spark plasma sintering (SPS) process [79]. Copyright Wiley and Sons, 2014.

**Figure 2 nanomaterials-09-00086-f002:**
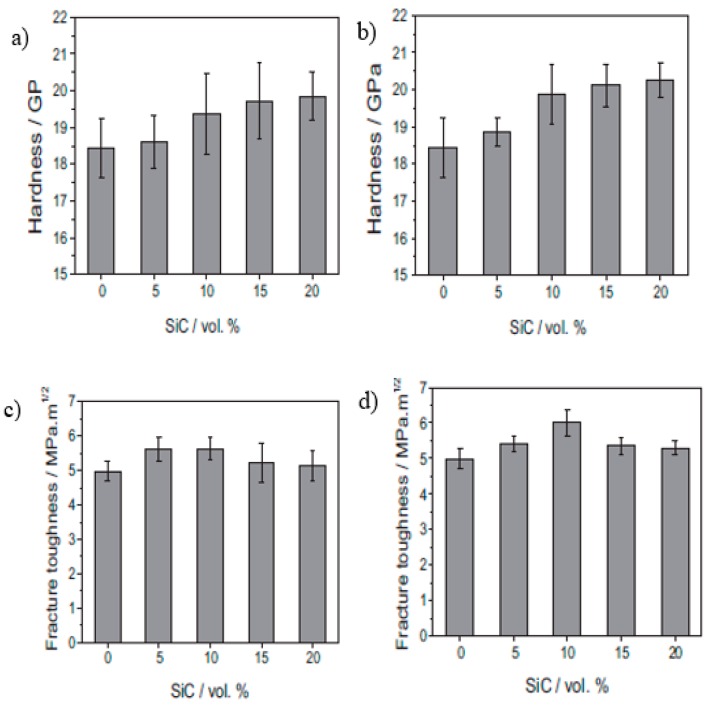
Mechanical properties of alumina–SiC nanocomposites: (**a**,**b**) hardness; (**c**,**d**) fracture toughness. Reproduced with permission from [10]. Copyright Elsevier, 2014.

**Figure 3 nanomaterials-09-00086-f003:**
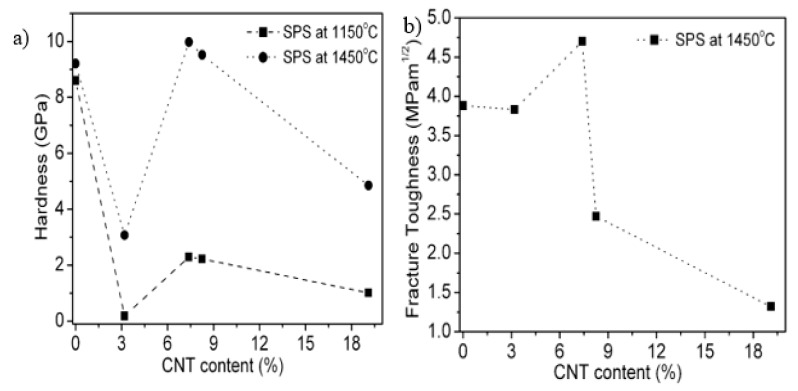
Mechanical hardness (**a**) and fracture toughness (**b**) variation with carbon nanotube (CNT) content. Reproduced with permission from [111]. Copyright Elsevier, 2009.

**Figure 4 nanomaterials-09-00086-f004:**
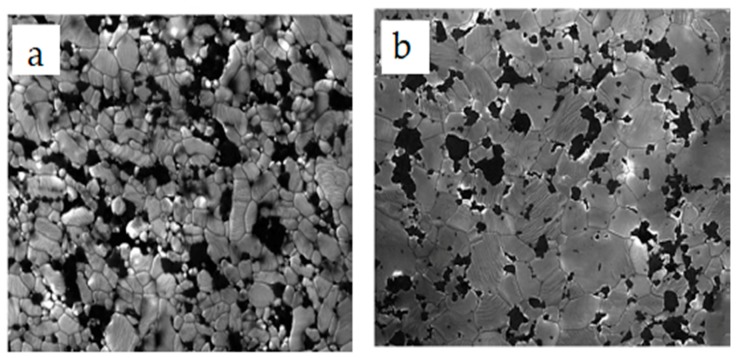
Field-emission scanning electron microscopy (FE–SEM) microstructure of Al_2_O_3_–SiC nanocomposites: (**a**) 1400 °C; (**b**) 1500 °C at 2 μm magnifications. Reproduced with permission from [86]. Copyright Elsevier, 2012. [86].

**Figure 5 nanomaterials-09-00086-f005:**
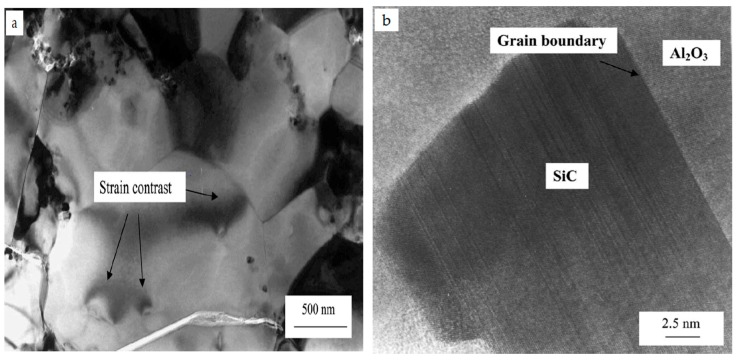
High-resolution transmission electron microscope (HRTEM) micrographs of Al_2_O_3_-5SiC nanocomposites: (**a**) distribution of SiC particles; (**b**) phase boundary between SiC and alumina. Reproduced with permission from [94]. Copyright Springer Nature, 2002.

**Figure 6 nanomaterials-09-00086-f006:**
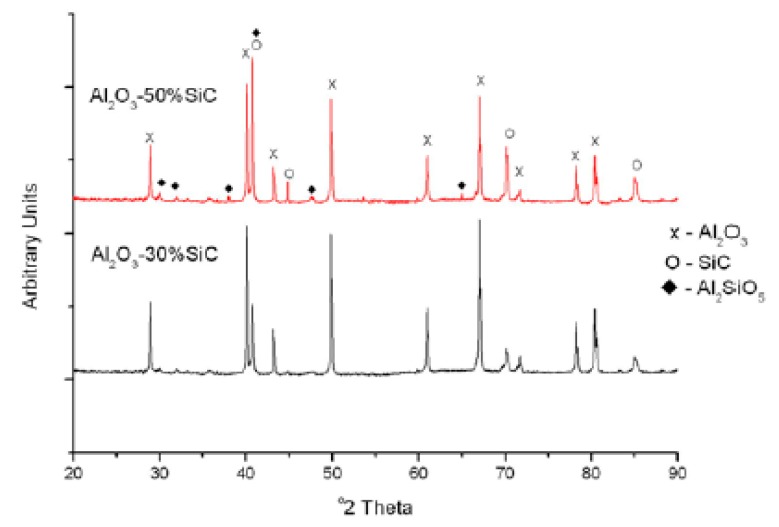
X-ray diffraction (XRD) of Al_2_O_3_–SiC nanocomposite. Reproduced with permission from [120]. Copyright International Association of Engineers, 2014.

**Figure 7 nanomaterials-09-00086-f007:**
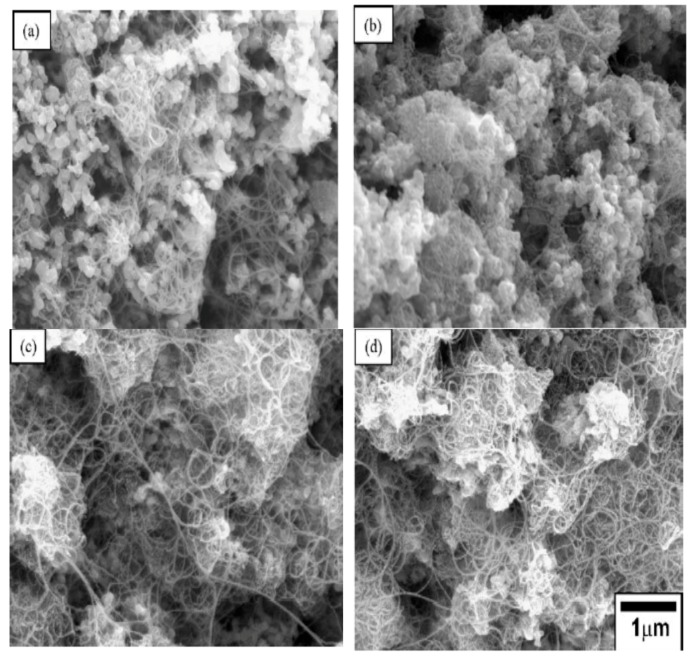
FE–SEM of Al_2_O_3_-CNT nanocomposites with varying CNT content (**a**) 7.39 wt%, (**b**) 8.25 wt%, (**c**) 18.82 wt% and (**d**) 19.10 wt%. Reproduced with permission from [156]. Copyright Elsevier, 2008.

**Figure 8 nanomaterials-09-00086-f008:**
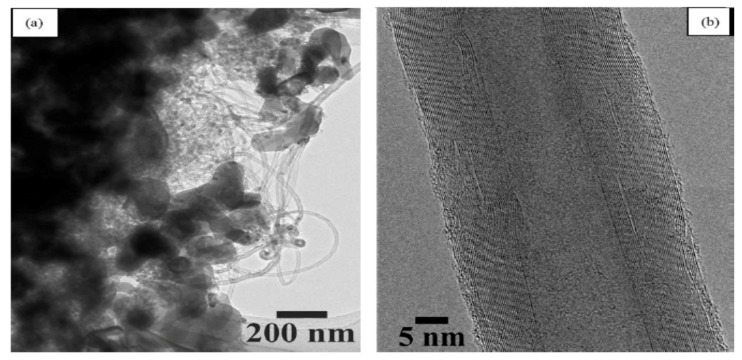
Transmission electron microscope (TEM) images of Al_2_O_3_-CNT nanocomposites at different resolutions (**a**) 200 nm (**b**) 5 nm. Reproduced with permission from [156]. Copyright Elsevier, 2008.

**Figure 9 nanomaterials-09-00086-f009:**
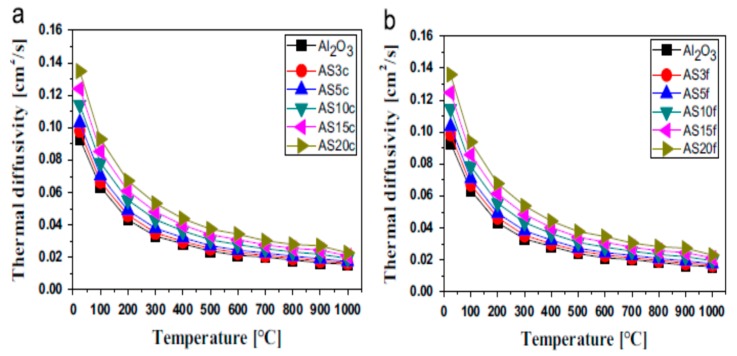
Thermal diffusivity vs measurement temperature: (**a**) nanocomposites AS*X*_c_; (**b**) composites AS*X*_f_ (f = fine, c = coarse). Reproduced with permission from [85]. Copyright Elsevier, 2014.

**Figure 10 nanomaterials-09-00086-f010:**
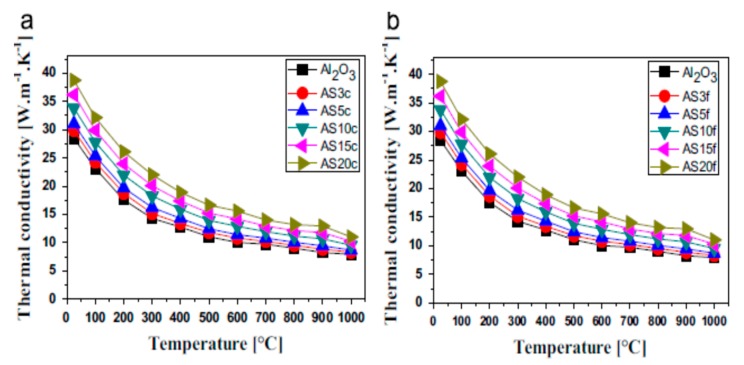
Thermal conductivity vs measurement temperature: (**a**) nanocomposites AS*X*_c_; (**b**) nanocomposites AS*X*_f._ (f = fine, c = coarse). Reproduced with permission from [85]. Copyright Elsevier, 2014.

**Figure 11 nanomaterials-09-00086-f011:**
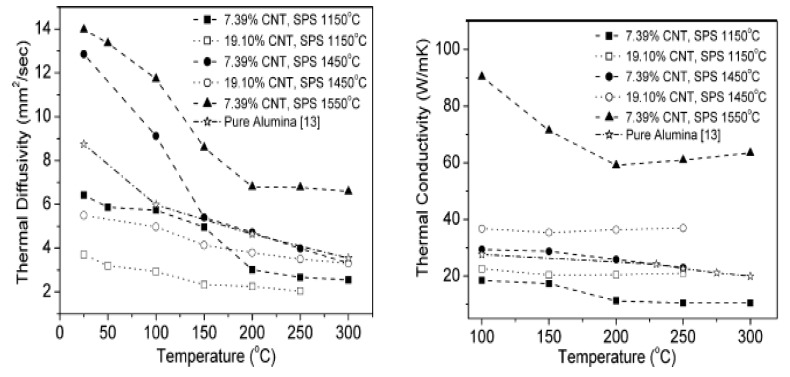
(**a**) Temperature dependence on thermal diffusivity; (**b**) variations of thermal conductivity with measurement temperature for Al_2_O_3_/CNT nanocomposites. Reproduced with permission from [112]. Copyright Elsevier, 2008.

**Figure 12 nanomaterials-09-00086-f012:**
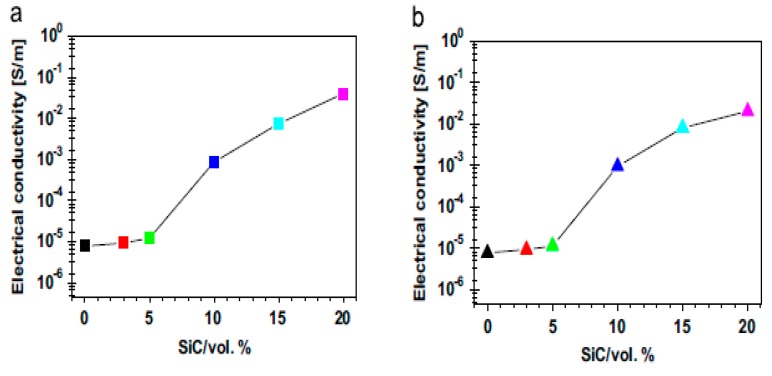
Electrical conductivity of Al_2_O_3_/SiC nanocomposites: (**a**) fine SiC; (**b**) coarse SiC. Reproduced with permission from [85]. Copyright Elsevier, 2014.

**Figure 13 nanomaterials-09-00086-f013:**
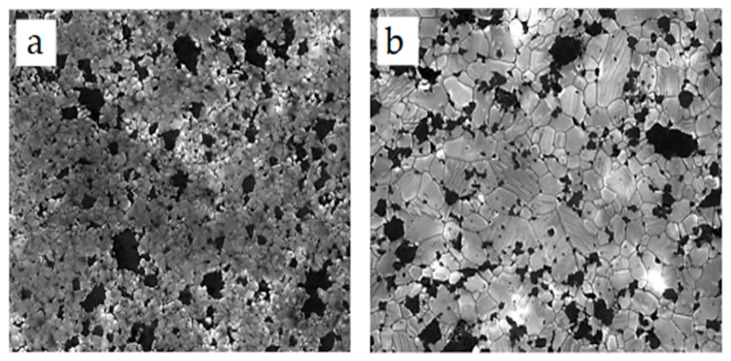
FE–SEM of Al_2_O_3_–SiC nanocomposites sintered by SPS: (**a**) 1400 °C; (**b**) 1550 °C at 2 μm magnification. Reproduced with permission from [86]. Copyright Elsevier, 2012.

**Table 1 nanomaterials-09-00086-t001:** Mechanical properties of Al_2_O_3_–SiC nanocomposites.

Nanocomposites	Consolidation Type	Densification (%)	Hardness (GPa)	Fracture Toughness (MPam^1/2^)	Strength (MPa)	Ref.
Al_2_O_3_–5SiC	SPS	99.5	19	4.5	980	[81]
Al_2_O_3_–5SiC	HP	-----	-----	4.7	467	[82]
Al_2_O_3_–17SiC	HP	98	22	4.6	383	[91]
Al_2_O_3_–5SiC	HP	99.9	20.4	3.3	760	[11]
Al_2_O_3_–5SiC	HP	98	18.8	-----	451	[138]
Al_2_O_3_–5SiC	SPS	98	-----	-----	380	[90]
Al_2_O_3_–5SiC	CIP	99.9	21.2	-----	-----	[83]
Al_2_O_3_–5SiC	HP	92	20.1	2.9	-----	[94]
Al_2_O_3_–5SiC	HP	98.2	24	7.1	363.8	[137]
Al_2_O_3_–10SiC	SPS	98.7	17.2	4.4	-----	[120]
Al_2_O_3_–10SiC	HP	99.1	22	5.4	550	[139]
Al_2_O_3_–5SiC	HP	99.67	-----	4.7	641	[122]
Al_2_O_3_–5SiC	SPS	99.76	21.78	2.65	-----	[133]

**Table 2 nanomaterials-09-00086-t002:** Mechanical properties of Alumina-CNT nanocomposites.

Nanocomposites	Consolidation Type	Densification (%)	Hardness (GPa)	Fracture Toughness (MPam^1/2^)	Strength (MPa)	Ref.
Al_2_O_3_–7.39MWCNT	SPS	79.1	9.98	4.7	-----	[111]
Al_2_O_3_–2MWCNT	HP	99	18	4.2	-----	[146]
Al_2_O_3_–10SWCNT	SPS	95.2	16.1	9.7	-----	[115]
Al_2_O_3_–4MWCNT	HP	98	17	4.2	380	[129]
Al_2_O_3_–0.1SWCNT	HP	95.28	15.55	2.85	-----	[130]
Al_2_O_3_–1MWCNT	SPS	-----	17	3.7	-----	[128]
Al_2_O_3_–4MWCNT	HP	-----	20	-----	-----	[43]
Al_2_O_3_–1MWCNT	HP	99.5	15.5	6.0	-----	[151]
Al_2_O_3_–0.3MWCNT	CIP	98.02	19.52	4.83	257.53	[149]
Al_2_O_3_–1MWCNT	Pressureless	99	17.1	4.1	543	[131]
Al_2_O_3_–0.9MWCNT	Pressureless	98.5	-----	5.83	742.6	[148]
Al_2_O_3_–3MWCNT	Pressureless	96.4	-----	4.7	363	[147]
Al_2_O_3_–0.3MWCNT	Pressureless	99.1	21	4.7	250	[152]
Al_2_O_3_–0.1MWCNT	SPS	-----	17.6	4.9	-----	[21]
Al_2_O_3_–0.5MWCNT	SPS	-----	15.0	5.8	420	[127]
Al_2_O_3_–1MWCNT	SPS	-----	-----	5.0	-----	[104]

**Table 3 nanomaterials-09-00086-t003:** Thermal conductivity of Al_2_O_3_–SiC nanocomposites.

Nanocomposites	Consolidation Type	Densification (%)	Thermal Conductivity (W/mK)	Ref.
Al_2_O_3_–20SiC	HP	99.3	38	[85]
Al_2_O_3_–60SiCw	HP	-----	42.1	[157]
Al_2_O_3_–30SiCp	HP	99	49	[169]
Al_2_O_3_–20SiCw	HP	98.3	34.25	[170]

**Table 4 nanomaterials-09-00086-t004:** Thermal conductivity of alumina–CNT nanocomposites.

Nanocomposites	Consolidation Type	Densification (%)	Thermal Conductivity (W/mK)	Ref.
Al_2_O_3_–10SWCNT	SPS	95.2	11.4	[115]
Al_2_O_3_–7.39MWCNT	SPS	84.2	90.44	[112]
Al_2_O_3_–0.15MWCNT	CIP	98.45	47.14	[149]

**Table 5 nanomaterials-09-00086-t005:** Electrical conductivity of Al_2_O_3_–SiC nanocomposites.

Nanocomposites	Consolidation Type	Densification (%)	Electrical Conductivity (S/m)	Ref.
Alumina–20SiC	HP	99.3	4.05 × 10^−2^	[85]
Alumina–17SiC	HP	99.0	5.88 × 10^−3^	[86]

**Table 6 nanomaterials-09-00086-t006:** Electrical conductivity of alumina–CNT nanocomposites.

Nanocomposites	Consolidation Type	Densification (%)	Electrical Conductivity (S/m)	Ref.
Alumina–3%MWCNT	SPS	-----	1.245	[188]
Alumina–5%MWCNT	SPS	99	576	[114]
Alumina–19.1%MWCNT	SPS	59.7	3336	[156]
Alumina–1%MWCNT	SPS	-----	2.5	[49]
Alumina–5.7%SWCNT	SPS	100	1050	[115]
Alumina–1%MWCNT	HP	99.5	10^−2^	[151]
Alumina–2.4%MWCNT	CIP	83.96	0.1	[149]

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
