# Peer review of "Recent Advances in the Processing and Properties of Alumina–CNT/SiC Nanocomposites"

_nanomaterials, 2019, doi:10.3390/nano9010086_

Reviewer 1 Report

The manuscript entitled "Recent advances in the processing and properties of alumina CNT / SiC nanocomposites" is a review about the formulation and the characterization of alumina-based nanocomposites with tailored properties. It is a very good paper and then I suggest its publication in Nanomaterials, after that the following issues were addressed:

Please, provide the definition of SiC in the abstract;

In my view Figures 1 and 2 reporting the pictures of the mechanical milling machines and sonicators respectively, are not useful;

Please, to enhance the readability of the paper, add some Figures in the part of the manuscript devoted to the discussion about the mechanical properties of the alumina-based nanocomposites;

Please, enhance the resolution of Figures 12.

Author Response

Response to Reviewer 1 Comments

Point 1: Please, provide the definition of SiC in the abstract

Response 1: The definition of SiC has been provided as suggested by the reviewer.

Point 2: In my view Figures 1 and 2 reporting the pictures of the mechanical milling machines and sonicators respectively, are not useful

Response 2: All the authors agreed to the reviewer’s opinion and therefore figure 1 and figure 2 reporting milling machines and sonicators respectively have been removed from the manuscript.

Point 3: Please, to enhance the readability of the paper, add some Figures in the part of the manuscript devoted to the discussion about the mechanical properties of the alumina-based nanocomposites.

Response 3:  Two figures (figure 2 and figure 3) have been added to the manuscript as suggested by the reviewer.

Point 4: Please, enhance the resolution of Figures 12.

Response 4: The resolution of figures 12 have been enhanced inline with reviewer suggestion.

 Reviewer 2 Report

Al-Aqeeli provide a good overview of nanocomposites based on alumina, carbon nanotubes and SiC. The review covers many aspects of the field and is well-written. I recommend publication after minor revisions.

(1) Please define a "nanocomposite" at the beginning and use consequently this term throughout the paper (do not exchange with such terms as "composite").

(2) The Chapter 7 on applications should be extended and moved to the first part of the review.

(3) Please indicate for all reproduced pictures if you obtained permission from their sources.

(4) There are still many editorial / occasional language mistakes. Please correct them. These include missing subscripts in formulae.

Author Response

Response to Reviewer 2 Comments

 Point 1:  Please define a "nanocomposite" at the beginning and use consequently this term throughout the paper (do not exchange with such terms as "composite").

 Response 1: The term nanocomposite has been defined as contained in the first paragraph of the manuscript. However, we have ascribed the term nanocomposites where it is applied as suggested by the reviewer.

 Point 2: The Chapter 7 on applications should be extended and moved to the first part of the review.

 Response 2: The authors agreed that the position of applications as placed in the manuscript is appropriate. Bringing it to the first part may confuse the readers.

 Point 3:  Please indicate for all reproduced pictures if you obtained permission from their sources.

 Response 3: The permission to reproduce the images and figures have been obtained from the appropriate sources.

 Point 4: There are still many editorial / occasional language mistakes. Please correct them. These include missing subscripts in formulae.

 Response 4: The authors agreed with the reviewer’s comment and they  have been fixed as contained in the manuscript.

 Round  2

Reviewer 1 Report

I recommend the publication of the manuscript in the present form, given that all the issues have been addressed by the authors.

Author Response

Many Thanks.

Reviewer 2 Report

Point 2:
extension of this chapter was also requested. This is not addressed.
Point 3:
there is no statement in the Figures captions. Please include in each reproduced figure caption a statement that permission was obtained.

Author Response

Point 2: The Chapter 7 on applications should be extended and moved to the first part of the review.

Response 2: The authors agreed that the position of applications as placed in the manuscript is appropriate. Bringing it to the first part of the manuscript may confuse the readers, however, the extension on the applications have been done as suggested by the reviewer and highlighted in red ink in the manuscript. 

Point 3:  Please indicate for all reproduced pictures if you obtained permission from their sources.

Response 3: The permission to reproduce the images and figures have been obtained from the appropriate sources. permission for each of the figures as contained in the manuscript were obtained and a sample of one of the permissions obtained from the respective paper containing the figures is attached to this response for your reference. Again, we have all the obtained license permissions as obtained from the various paper sources containing the figures and saved for future reference.  

Round  3

Reviewer 2 Report

The major points are addressed. The figures captions are still not changed - decision left to the Editor.